# The [simple carbon project] model v1.0

Cameron O'Neill[1], Andrew McC. Hogg[1], Michael Ellwood[1], Stephen Eggins[1], and Bradley Opdyke[1]

[1]Research School of Earth Sciences, Australian National University

*Correspondence to:* Cameron O'Neill (cameron.oneill@anu.edu.au)

**Abstract.** We construct a carbon cycle box model to process observed or inferred geochemical evidence from modern and paleo settings. The [simple carbon project] model v1.0 ("SCP-M") combines a modern understanding of the ocean circulation regime with the earth's carbon cycle. SCP-M estimates the concentrations of a range of elements within the carbon cycle, by simulating ocean circulation, biological, chemical and atmospheric and terrestrial carbon cycle processes. The model is capable of reproducing both paleo and modern observations, and aligns with CMIP5 model projections. SCP-M's fast run time, simplified layout and matrix structure render it a flexible and easy-to-use tool for paleo- and modern carbon cycle simulations. The ease of data integration also enables model-data optimisations. Limitations of the model include the prescription of many fluxes, and an ocean basin-averaged topology, which may not be applicable to more detailed simulations.

In this paper we demonstrate SCP-M's application primarily with analysis of the Last Glacial Maximum (LGM) to Holocene carbon cycle transition, and also with the modern carbon cycle under the influence of anthropogenic $CO_2$ emissions. We conduct an atmospheric and ocean multi-proxy model-data parameter optimisation for the LGM and late Holocene periods, using the growing pool of published paleo atmosphere and ocean data for $CO_2$, $\delta^{13}C$, $\Delta^{14}C$ and carbonate ion proxy. The results provide strong evidence for an ocean-wide physical mechanism to deliver the LGM to Holocene carbon cycle transition. Alongside ancillary changes in ocean temperature, volume, salinity, sea ice cover and atmospheric radiocarbon production rate, changes in global overturning circulation, and, to a lesser extent Atlantic meridional overturning circulation, can drive the observed LGM and late Holocene signals in atmospheric $CO_2$, $\delta^{13}C$, $\Delta^{14}C$, and the oceanic distribution of $\delta^{13}C$, $\Delta^{14}C$ and carbonate ion proxy. Further work is needed on analysis and processing of the ocean proxy data to improve confidence in these modelling results.

## 1 Introduction

A box model divides regions of the ocean into boxes or grids, based on some property of the composite water masses, such as temperature, density or chemical composition. The model equations describe the evolution of tracers in the model's boxes, due to the various fluxes between each box (Fig. 1). Box models differ from more complex models such as General Circulation Models (GCM), mainly due to their reduced spatial resolution (i.e. much larger grids or boxes), and with major processes and fluxes typically prescribed rather than calculated in the model. Box models range in complexity from simple, basin-averaged models (e.g. Stommel, 1961; Sarmiento and Toggweiler, 1984; Toggweiler, 1999) to more complex, multi-basin ocean models (Hain et al., 2010) and full Earth carbon cycle models (Zeebe, 2012). Box models, despite being simpler than GCMs, have been

useful in illustrating key concepts in oceanography. For example, Sarmiento and Toggweiler (1984), Siegenthaler and Wenk (1984) and Knox and McElroy (1984) used simple four-box ocean-atmosphere models to show that the LGM $CO_2$ drawdown could have resulted from increased biological productivity and/or reduced ocean overturning circulation. More recently, Hain et al. (2010) used a box model to show that a range of ocean physical and biological mechanisms were required to cause

lower atmospheric $CO_2$ levels in the LGM, and Zhao et al. (2017) used a similar model to explore ocean ventilation ages in the LGM-deglacial and Holocene periods. Despite the development of highly complex coupled atmosphere-ocean models for climate simulations, box models continue to be applied to resolve problems in the carbon cycle.

Our motivation in constructing a new box model of the full carbon cycle, the [simple carbon project] model v1.0 ("SCP-M"), is to contribute a simple, easy to use, open access model implemented with freely available software, that is consistent with

physical and biogeochemical oceanography, that caters for important features of the carbon cycle, and that has explicit avenues for data integration, optimisation and inversion. Recent advances in physical oceanography have refined our understanding of global ocean circulation and mixing fluxes. For example, Talley (2013) provided a simplified interpretation of the global ocean as a handful of large-scale processes, some of which are operating across all basins - as is the case with the global overturning circulation. De Boer and Hogg (2014) described a simple model of deep ocean mixing of water masses under the influence of

seafloor topography. These high level concepts are easy to apply to box models. Furthermore, the growing pool of paleo proxy data across carbon isotopes and reconstructions (e.g. carbonate ion) presents an opportunity to progress model-data integrations using a number of different proxies. SCP-M caters for a range of proxies including the carbon isotopes and carbonate ion proxy, with the capacity for additional elements with minimal programming effort. The model-data experiment described in this paper provides a direct linkage between paleo-data and discrete values for ocean parameters in the LGM and late Holocene periods,

thus contributing our understanding of the LGM-Holocene carbon cycle transition. Combined with the expanding dataset of paleo observations, and with advances in computing power, data-aligned models such as SCP-M have the potential to improve our understanding of past changes in climate across many other timeframes. Furthermore, there are a number of features of the carbon cycle outside ocean circulation and biology, which influence proxy indicators, particularly the carbon isotopes. Omitting these features could lead to erroneous modelling outcomes, particularly in the case of the terrestrial biosphere which strongly

influences atmospheric $CO_2$ and $\delta^{13}C$ (e.g. Francois et al., 1999). We compiled SCP-M to include a broad range of carbon cycle mechanisms, including carbonate production and dissolution, marine sediments, terrestrial biosphere, anthropogenic emissions sources and continental weathering. Finally, SCP-M is computationally cheap and quick to run. A 10,000 year simulation takes approximately thirty seconds to process on a regular laptop, enabling exhaustive exploration of parameter space in optimisations that incorporate large datasets. While box models are not new, we argue that these features contribute

to a new tool that is well-equipped to tackle a wide range of applications in paleoceanography, paleo-climate and the modern carbon cycle.

In this paper we describe SCP-M and illustrate its application alongside LGM and late Holocene period ocean and atmosphere data, with several insights for the transition between the two periods, plus modelling of the modern and future carbon cycle under the influence of anthropogenic emissions. Emphasis is placed on the model description and configuration.

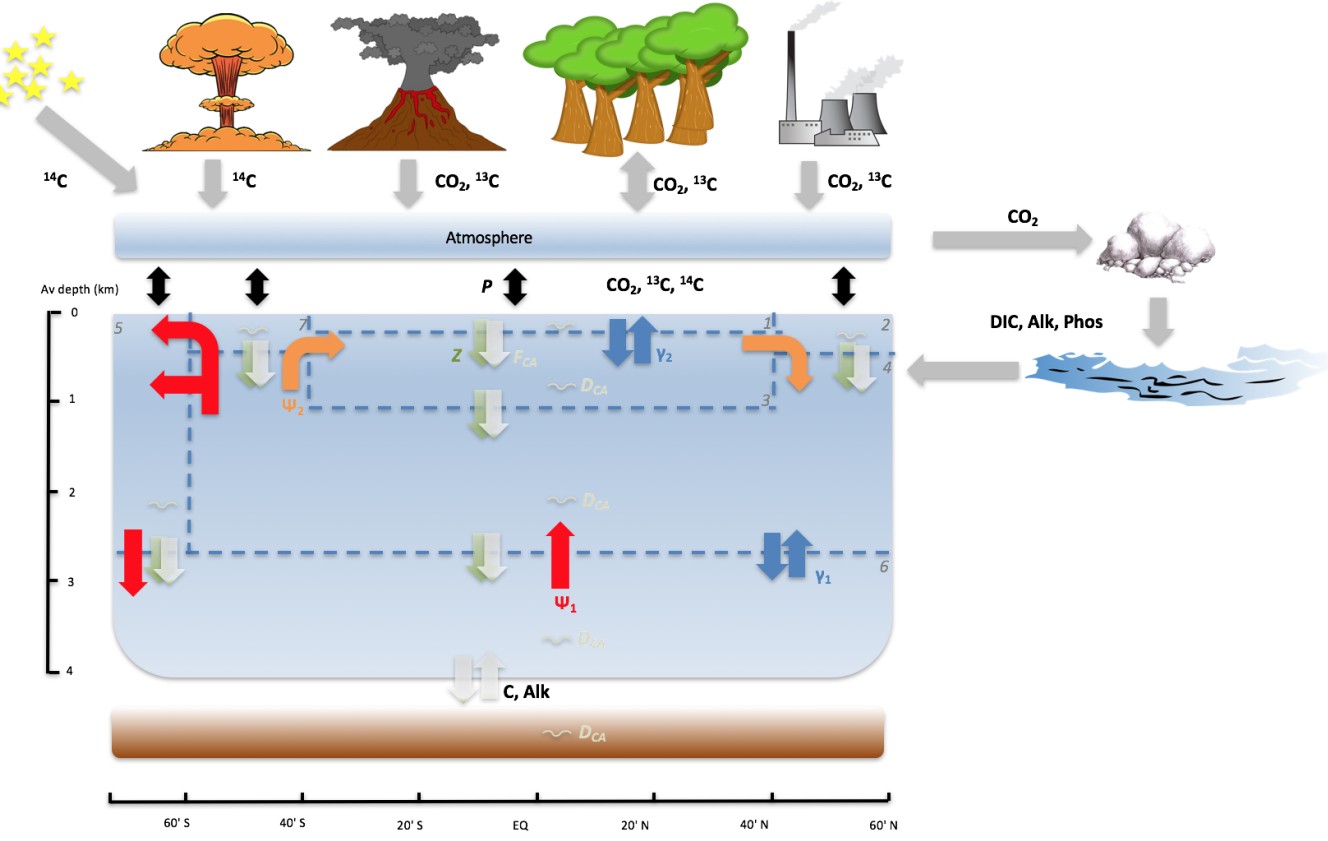

**Figure 1.** SCP-M: configured as a seven box ocean model-plus atmosphere with marine sediments, continents and the terrestrial biosphere. Exchange of elemental concentrations, e.g. $C_i, (i = 1, 7)$ occur due to fluxes between boxes. $\Psi_1$ (red arrows) is global overturning circulation (GOC), $\Psi_2$ (orange arrows) is Atlantic meridional overturning circulation (AMOC), $\gamma_1$ (blue arrows between boxes 4 and 6) is deep-abyssal mixing, $\gamma_2$ (blue arrows between boxes 1 and 3) is low-latitude thermohaline mixing, $Z$ (green downward arrows) is the biological pump, $F_{CA}$ (white downward arrows) is the carbonate pump, $D_{CA}$ (white squiggles) is carbonate dissolution and $P$ (black, bidirectional arrows) is the air-sea gas exchange. Box 1: low latitude/tropical surface ocean; box 2: northern surface ocean; box 3: intermediate ocean; box 4: deep ocean; box 5: Southern Ocean; box 6: abyssal ocean; box 7: subpolar southern surface ocean.

## 2 SCP-M description

SCP-M is focussed on the ocean carbon cycle and is configured to estimate the time evolution of oceanic dissolved inorganic carbon (DIC) and its constituents, $\delta^{13}C$, $\Delta^{14}C$, plus alkalinity, phosphorus, oxygen and atmospheric $CO_2$, $\delta^{13}C$, and $\Delta^{14}C$. It contains a simple, yet realistic representation of large scale ocean physical processes, with an overlay of ocean chemistry and biology (Fig. 1). SCP-M simulates sources and sinks of carbon across the ocean and atmosphere, marine sediments and terrestrial biosphere. Volcanic emissions, sedimentary weathering, rivers and anthropogenic emissions are prescribed fluxes. A

range of carbon cycle features are included, because the concentration of carbon in the ocean and atmosphere (and its isotopes in particular) are sensitive to many sources and sinks, and omitting them makes it difficult to compare model results with the carbon data that indelibly features their imprint. For example, regrowth in the terrestrial biosphere imparts a clear signature on the atmosphere and ocean $\delta^{13}$C data profile after the LGM (e.g. Francois et al., 1999; Ciais et al., 2012; Hoogakker et al., 2016). In addition, the atmospheric radiocarbon source, marine sediments, volcanic emissions, continental weathering, and now anthropogenic emissions, exert important influences on carbon cycle observations.

SCP-M was designed to compare model results with data, and to solve for optimal parameter combinations. Within SCP-M, realistic implementation of physical processes upon a sound biogeochemical platform enables their transmission into paleo-chemical tracer signals, for which proxy data exists. Many of the key ocean physical and biological processes are prescribed in the model, allowing them to be free parameters in model-data experiments. SCP-M itself is implemented with a matrix framework which enables more boxes to be added, ocean basins to be separated, elements to be added, exploration of a range of hypotheses, all with minimal programming effort.

## 2.1 Model topology

The box model is mostly conceptual in nature and is designed to test high-level concepts. Therefore, excessive detail and complication is to be avoided. However, key processes that are critical to the validity of any hypothesis being tested, must be represented as well as possible.

The ocean is a key part of the global carbon cycle and pre-eminent in hypotheses of glacial/interglacial carbon cycles (e.g. Kohfeld and Ridgewell, 2009; Sigman et al., 2010). Talley (2013) provided an observationally-based description of ocean circulation in terms of its constituent water masses, circulation and mixing fluxes, and including estimates of the present day magnitudes of those fluxes. The Talley (2013) model builds on the models of Broecker (1991), Gordon (1991), Schmitz (1996), Lumpkin and Speer (2007), Kuhlbrodt et al. (2007), Talley (2008), and Marshall and Speer (2012). Key features of the Talley (2013) model include:

– Atlantic thermocline water moves north ultimately reaching the North Atlantic, driven by advection and surface buoyancy changes. High salinity North Atlantic Deep Water (NADW) forms in the north by cooling, densification and convection, and then travels south to rise up into the Southern Ocean via wind-driven upwelling and Ekman flows, forming Lower Circumpolar Deep Water (LCDW). This water comprises the upper (orange arrows) Atlantic meridional overturning circulation (AMOC) in SCP-M (Fig. 1).

– A fraction of the upwelled LCDW sinks to become Antarctic Bottom Water (AABW) under the influence of cooling and brine rejection, south of the Antarctic Circumpolar Current (ACC). AABW moves northward along the ocean floor via adiabatic advection (Talley, 2013) in all basins. It upwells into deep water in the Pacific and Indian Oceans and also into NADW in the Atlantic via upwelling with diapycnal diffusion (Talley, 2013). The combined LCDW/AABW/PDW/IDW/NADW global overturning circulation (GOC) is represented by the red arrows in Fig. 1.

- Pacific Deep Water/Indian Deep Water (PDW/IDW) upwells at low latitudes and returns to the Southern Ocean above the NADW, forming the core of the Upper Circumpolar Deep Water (UCDW), which is identified by Talley (2013) as low oxygen content (old) water. A part of the upwelled PDW/IDW joins AABW formation, with the bulk of it moving north from the Southern Ocean at shallow and intermediate depths. These waters become fresher and warmer, and join Antarctic Intermediate Water (AAIW) and Subantarctic Mode Water (SAMW) at the base of the subtropical thermocline (advection with surface buoyancy fluxes).

- Joined thermocline waters, AAIW/SAMW and upwelled thermocline waters from the Pacific and Indian Oceans, form the upper ocean transport moving towards the North Atlantic.

A key contribution of the Talley (2013) study is that GOC is the pre-eminent process in distributing water throughout the global oceans. Talley (2013) provided a 2-D 'collapsed' interpretation of a 3-D ocean layout, based on the observation that similar, large scale processes (i.e. GOC) operate in all three major ocean basins, and this interpretation can directly inform a box model topology. The Talley (2013) 2-D global ocean view, used in SCP-M, captures the features described above in a simple ocean box model format. Talley (2013) also provided observation-based estimates of the ocean transport fluxes, which are scaled according to their ocean basin domain. For example, the GOC and AABW-formation process is common to all basins, and thus accounts for the largest flux, of 29 Sverdrups (Sv, $10^6$ m$^3$ s$^{-1}$). The AMOC/NADW sinking cell is confined to the Atlantic Basin and represents a smaller flux (19 Sv) of water (Talley, 2013).

The SCP-M dimensions are designed to be consistent with measured estimates of the ocean surface area and average depth, and total ocean and atmosphere volumes. The model is divided into boxes according to latitude and depth, but not by longitude. Therefore, it does not distinguish between Atlantic, Pacific and Indian Basins, and does not allow for compositional variations with longitude. Each box has a surface area, depth (and therefore volume), and corresponds to a location in the global ocean with reference to latitude and average depth. It is simple to add more boxes to divide the model into ocean basins.

SCP-M contains seven ocean boxes as shown in Fig. 1. The rationale for dividing the ocean into boxes is that there are regions of the ocean that are relatively well mixed, or at least similar in terms of their prevailing element flux behaviour. For the depth of the surface boxes, this rationale conveniently translates to the maximum wintertime mixed layer depth (MLD) (e.g. Kara et al., 2003; de Boyer Montegut et al., 2004). We choose a depth of 100m for Box 1, the low latitude surface box, which is a reasonable approximation to the 20-150m seasonally-varying MLD for the mid and low latitudes estimated by de Boyer Montegut et al. (2004), and consistent with the depth of a similar box in the Toggweiler (1999) model. This box represents the photic zone over much of the ocean, from 40°S to 40°N. Craig (1957) estimated the depth of this layer as 75m ±25m, a value used by Keeling and Bolin (1968) in their simple ocean box model. We choose 250m depth for the NADW box (box 2) and the subpolar surface box (box 7) as per Toggweiler (1999). These boxes are deeper than the low latitude surface box (de Boyer Montegut et al., 2004), in order to capture the regions of deep water upwelling (subpolar Southern Ocean) and convective downwelling (North Atlantic). The MLD in these regions can vary between 70 and >500m depth depending on seasonal variations (de Boyer Montegut et al., 2004). An intermediate depth box (3) resides below the low latitude surface box and extends from 100m depth to 1000m depth. This box captures northward flowing AAIW and SAMW from upwelled

NADW/PDW/IDW (e.g. Talley, 2013). Box 4 is the deep ocean box, extending from 1,000m depth to 2,500m depth and incorporates the upwelling abyssal waters in all basins, and downwelled NADW. This water is channeled back to the surface in the subpolar surface box and the Southern Ocean box, as per the wind-driven upwelling of Morrison and Hogg (2013) and Talley (2013). The Southern Ocean box (5) extends from 80°S to 60°S and from the ocean surface to 2,500m depth. This box encompasses the Southern Ocean, the ACC and deep water formation from southward flowing upwelled NADW/PDW/IDW (Talley, 2013). The abyssal box (6) extends the full range of the ocean, from 2,500m to 4,000m depth (our assumed average depth of the ocean). This box is the pathway for northward flowing AABW and incorporates mixing with overlying deep water and advection/upwelling (Talley, 2013).

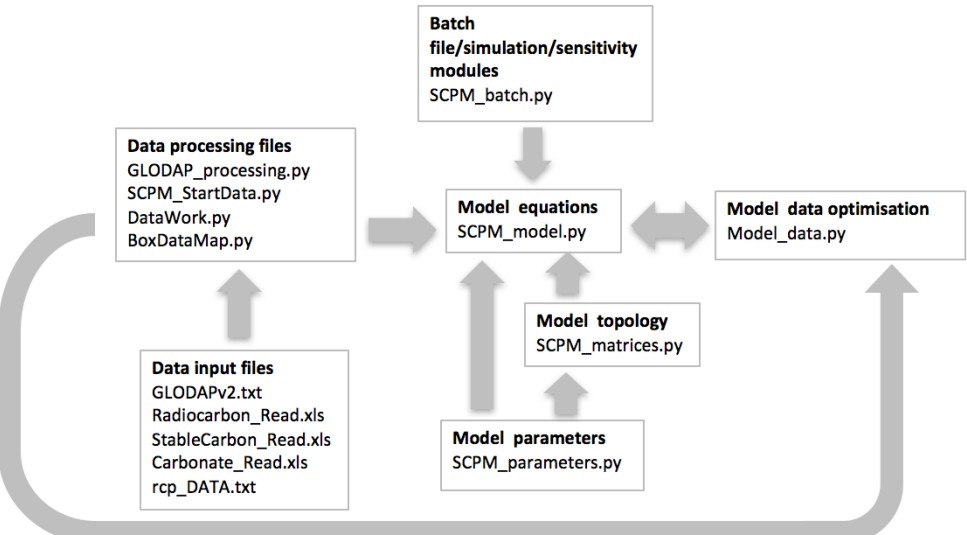

**Figure 2.** SCP-M Python and ancillary files with their linkages. Arrows refer to the direction of file linkages and the order of their activation during the routine of setting up and running the model. SCP-M is currently implemented in Python 3.6, although has been run on other versions of Python. Folder/file structure separates model and data files. All files and user manual are available from (https://doi.org/10.5281/zenodo.1310161).

## 2.2 The model parameters, processes and equations

### 2.2.1 Basic features

Figure 2 shows the suite of files used to execute SCP-M. We have chosen a modular approach to reduce complexity of each of the model files. The SCP-M routine includes data processing for the model's boxes based on geographic coordinates, model calibration to the data, model simulations, model-data optimisation and charting/tabular output. SCP-M is implemented in Python 3.6, with the code and download/user instructions available at (https://doi.org/10.5281/zenodo.1310161).

In short, SCP-M calculates the evolution of an element's or species' concentration in each model box, as a function of time and flux parameters (e.g. inputs and outputs to each box), or processes, such as uptake or regeneration. The model includes ocean circulation and mixing fluxes, air-sea gas exchange, chemical and biological transformations, and sources and sinks of carbon. The model equations are a set of partial differential equations, one for each element in the model. These are solved with a straightforward, first order Euler forward time-stepping method with a standard time step of one year. We find the model to be stable and approaching steady state for most of the simulations that we undertook. However, this stability can be challenged by scenarios with strong forcing. With the Euler method, errors can propagate in proportion to the step size. This can be resolved either by revising the selection of parameter inputs or starting data values, or by reducing the size of the time step in each model run.

## 2.2.2 The ocean circulation and mixing

There are four ocean physical parameters in SCP-M. $\Psi_1$ and $\Psi_2$ are advection terms that represent the physical transport of water from one box to another, containing the element or species concentration of its box of origin. $\Psi_1$ represents GOC (e.g. Sarmiento and Toggweiler, 1984; Marshall and Speer, 2012; Talley, 2013) that infiltrates all basins (Talley, 2013) and is shown by the red arrows in Fig. 1. The $\Psi_1$ parameter allows a variable allocation between transport from the deep ocean box (box 4) into the subpolar surface box (box 7) and directly into the polar box (box 5), via $\alpha$. $\Psi_2$ represents the AMOC. This is the region of northward-flowing intermediate waters in the Atlantic Ocean, and formation of NADW (Dickson and Brown, 1994; Talley, 2013), shown as orange arrows in Fig. 1. $\gamma_1$ and $\gamma_2$ are bidirectional mixing terms that exchange element or species concentrations between boxes without any net advection of water (blue arrows in Fig. 1). $\gamma_1$ is bidirectional mixing between the deep and abyssal boxes of the form described by Lund et al. (2011) and De Boer and Hogg (2014). $\gamma_2$ is a low latitude, intermediate-shallow box "thermocline" mixing parameter, which governs the constant bidirectional exchange between these two boxes (Liu et al., 2016).

The influence of each of the ocean parameters is prescribed in box model space by matrix equations, with one matrix for each parameter. Each row and column position in the matrix corresponds to a box location. The atmosphere box is treated separately from the ocean boxes, and it does not enter the ocean parameter matrices. The volumetric circulation or mixing parameters, in Sv, are multiplied by the oceanic element concentration (mol m$^{-3}$) to produce a molar flux exchanged between ocean boxes. For example the change in concentration of carbon (as DIC) in the deep box (box 4) from ocean physical parameters is estimated by:

$$\left[\frac{dC_4}{dt}\right]_{phys} = \frac{\Psi_1(C_6 - C_4)}{V_4} + \frac{\Psi_2(C_2 - C_4)}{V_4} + \frac{\gamma_1(C_6 - C_4)}{V_4} \tag{1}$$

where $C_i$ is the concentration of DIC in each box in mol m$^{-3}$ and $V_i$ is the volume of each box in m$^3$. There is no vertical flux between box 4 and box 3 (intermediate box) in Eq. (1). We assume that this vertical flux is small compared with the lateral transport, and compared with the mixing fluxes between boxes 4 and 6 (and boxes 1 and 3 in Eq. 2 below). We assume that the boundary between boxes 3 and 4 is the divide between northward flowing water sourced from AAIW and SAMW, which overlies southward return flow from AMOC and PDW/IDW. The fluxes out of box 4 are shown by the terms $-\Psi_1 C_4$, $-\Psi_2 C_4$

and $-\gamma_1 C_4$, with the fluxes into boxes 5, 6 and 7 treated in the equations for those boxes. For the low latitude surface box (box 1):

$$\left[\frac{dC_1}{dt}\right]_{phys} = \frac{\gamma_2(C_3 - C_1)}{V_1} \tag{2}$$

We assume that box 1 represents the shallow, mixed layer of the ocean (e.g. Kara et al., 2003; de Boyer Montegut et al.,
2004), which is mainly influenced by surface processes. We prescribe vertical mixing between this box and the underlying intermediate box (3) via the $\gamma_2$ parameter. This represents the thermocline mixing described by Liu et al. (2016). We assume that lateral transport of northward flowing water underlies box 1, involving box 7 (subpolar Southern Ocean), box 3 and box 1 (Northern ocean). This intermediate depth water is colder and denser than the overlying mixed layer, given its provenance of AAIW, SAMW and deep-upwelled NADW/PDW/IDW (Talley, 2013). These ocean circulation and mixing operations (e.g.
Eq. 1 and Eq. 2) can be vectorised for all boxes using sparse matrices, as follows:

$$\left[\frac{d\mathbf{C}}{dt}\right]_{phys} = \frac{(\Psi_1 \mathsf{T}_1 + \Psi_2 \mathsf{T}_2 + \gamma_1 \mathsf{E}_1 + \gamma_2 \mathsf{E}_2) \cdot \mathbf{C}}{\mathbf{V}} \tag{3}$$

where:

$$\mathbf{C} = C_i, \; for \; i = 1, 7 \tag{4}$$

$$\mathbf{V} = V_i, \; for \; i = 1, 7 \tag{5}$$

and $\mathsf{T}_1$, $\mathsf{T}_2$, $\mathsf{E}_1$ and $\mathsf{E}_2$ are sparse matrices defined as:

$$\mathsf{T}_1 = \begin{pmatrix} 0 & 0 & 0 & 0 & 0 & 0 & 0 \\ 0 & 0 & 0 & 0 & 0 & 0 & 0 \\ 0 & 0 & 0 & 0 & 0 & 0 & 0 \\ 0 & 0 & 0 & -1 & 0 & 1 & 0 \\ 0 & 0 & 0 & (1-\alpha) & -1 & 0 & \alpha \\ 0 & 0 & 0 & 0 & 1 & -1 & 0 \\ 0 & 0 & 0 & \alpha & 0 & 0 & -\alpha \end{pmatrix} \tag{6}$$

$$\mathsf{T}_2 = \begin{pmatrix} 0 & 0 & 0 & 0 & 0 & 0 & 0 \\ 0 & -1 & 1 & 0 & 0 & 0 & 0 \\ 0 & 0 & -1 & 0 & 0 & 0 & 1 \\ 0 & 1 & 0 & -1 & 0 & 0 & 0 \\ 0 & 0 & 0 & 0 & 0 & 0 & 0 \\ 0 & 0 & 0 & 0 & 0 & 0 & 0 \\ 0 & 0 & 0 & 1 & 0 & 0 & -1 \end{pmatrix} \tag{7}$$

$$E_1 = \begin{pmatrix} 0 & 0 & 0 & 0 & 0 & 0 & 0 \\ 0 & 0 & 0 & 0 & 0 & 0 & 0 \\ 0 & 0 & 0 & 0 & 0 & 0 & 0 \\ 0 & 0 & 0 & -1 & 0 & 1 & 0 \\ 0 & 0 & 0 & 0 & 0 & 0 & 0 \\ 0 & 0 & 0 & 1 & 0 & -1 & 0 \\ 0 & 0 & 0 & 0 & 0 & 0 & 0 \end{pmatrix} \tag{8}$$

$$E_2 = \begin{pmatrix} -1 & 0 & 1 & 0 & 0 & 0 & 0 \\ 0 & 0 & 0 & 0 & 0 & 0 & 0 \\ 1 & 0 & -1 & 0 & 0 & 0 & 0 \\ 0 & 0 & 0 & 0 & 0 & 0 & 0 \\ 0 & 0 & 0 & 0 & 0 & 0 & 0 \\ 0 & 0 & 0 & 0 & 0 & 0 & 0 \\ 0 & 0 & 0 & 0 & 0 & 0 & 0 \end{pmatrix} \tag{9}$$

Given we have applied the global ocean interpretation of Talley (2013) to the SCP-M layout, we have also adopted the observationally-based estimates for the large scale ocean fluxes for the modern ocean, from the same study: GOC ($\Psi_1$, 29 Sv), AMOC ($\Psi_2$, 19 Sv) and deep-abyssal mixing ($\gamma_1$, 19 Sv). For thermocline mixing between boxes 1 and 3 ($\gamma_2$), we have adopted the value for the corresponding flux from Toggweiler (1999), of 40 Sv.

### 2.2.3 Biological flux parameterisation

The biological pump (e.g. Broecker, 1982) is a descriptor of marine biological activity, whereby organisms consume nutrients in shallow waters, die, sink and then release those nutrients at depth. For example, carbon is taken up by shallow water-dwelling phytoplankton through photosynthesis and then sequestered in deeper waters after sinking, breaking down and re-mineralising their nutrient load back into the water column. Volk and Hoffert (1985) made the distinction between the soft tissue pump (STP), for soft tissued organisms, and the carbonate pump (carbonate-shelled organisms). We also distinguish between the two pumps, as they have different effects on carbon and alkalinity balances and therefore pCO$_2$ and carbonate dissolution. This section deals with the STP, and a following section deals with the carbonate pump. Most STP organic matter is remineralised in the shallow to intermediate ocean depths, leading to a decrease in the export of carbon as depth increases. According to Henson et al. (2011), only ∼15-25 per cent of organic material is exported to >100m depth, with most recycled in the shallower waters.

Martin et al. (1987) modelled the soft-bodied organic flux of carbon observed from sediment traps in the northeast Pacific to create a simple power rule which is easily applicable to modelling. The Martin et al. (1987) equation produces a flux of organic carbon, which is a function of depth from a base organic flux at 100m depth (the "Martin reference depth"). The flux at 100m

depth was estimated by Martin et al. (1987) to be between 1.2 and 7.1 mol C m$^{-2}$ yr$^{-1}$ from eight station observations in the northeast Pacific Ocean. Sarmiento and Gruber (2006) estimated a range of 0.0-5.0 mol C m$^{-2}$ yr$^{-1}$, with some localised higher values, across the global ocean. Equation (10) shows the general form of the Martin et al. (1987) equation:

$$F = F_{100}\left(\frac{z}{100}\right)^b \tag{10}$$

Where $F$ is a flux of carbon in mol C m$^{-2}$ yr$^{-1}$, $F_{100}$ is an estimate of carbon flux at 100m depth, $z$ is depth in metres and $b$ is a depth scalar. In SCP-M, the $Z$ parameter implements the Martin et al. (1987) equation. $Z$ is an estimate of biological productivity at 100m depth (in mol C m$^{-2}$ yr$^{-1}$), and coupled with the Martin et al. (1987) depth scalar, controls the amount of organic carbon that sinks from each model surface box to the boxes below. Each subsurface ocean box receives a flux of carbon from the box above it, at its ceiling depth (also the floor of the overlying box), and loses carbon as a function of the

depth of the bottom of the box. Remineralisation in each box is accounted for as the difference between the influx and out-flux of organic carbon. The biological flux out of the surface box 1 is shown by:

$$\left[\frac{dC_1}{dt}\right]_{bio} = \frac{Z_1 S_1 \left(\frac{d_{f1}}{d_0}\right)^b}{V_1} \tag{11}$$

where $Z_1$ is the biological flux of carbon prescribed for the surface box 1 in mol C m$^{-2}$ yr$^{-1}$, $S_1$ is the surface area of the surface box 1, $d_0$ is the reference depth of 100m for the $Z$ parameter value (Martin et al., 1987) and $d_c$ and $d_f$ are the ceiling

and floor depths of a box, respectively. The dimensionless parameter $b$ is the depth power function of the Martin et al. (1987) equation, which tapers biological production and export below depths of 100m. The net biological flux for intermediate depth Box 3 is given by:

$$\left[\frac{dC_3}{dt}\right]_{bio} = \frac{Z_1 S_1 \left[\left(\frac{d_{c3}}{d_0}\right)^b - \left(\frac{d_{f3}}{d_0}\right)^b\right]}{V_3} \tag{12}$$

The process is vectorised using sparse matrices in the following:

$$\left[\frac{d\mathbf{C}}{dt}\right]_{bio} = \frac{\mathbf{ZS} \cdot (\mathsf{B_{out}} + \mathsf{B_{in}})}{\mathbf{V}} \tag{13}$$

where $\mathbf{Z}$ is an array of the $Z_i$ (i=1,7) parameter which varies across the surface boxes and $\mathbf{S}$ is the array of surface box surface areas $S_i$ (i=1,7). As with the ocean parameters, the biological flux of carbon is divided through by the box volume array $\mathbf{V}$ to return concentrations in mol m$^{-3}$. $\mathsf{B_{out}}$ and $\mathsf{B_{in}}$ are sparse matrices as follows:

$$\mathsf{B_{out}} = \begin{pmatrix} -\left(\frac{d_{f1}}{d_0}\right)^{-b} & 0 & 0 & 0 & 0 & 0 & 0 \\ 0 & -\left(\frac{d_{f2}}{d_0}\right)^{-b} & 0 & 0 & 0 & 0 & 0 \\ -\left(\frac{d_{f3}}{d_0}\right)^{-b} & 0 & 0 & 0 & 0 & 0 & 0 \\ -\left(\frac{d_{f4}}{d_0}\right)^{-b} & -\left(\frac{d_{f4}}{d_0}\right)^{-b} & 0 & 0 & 0 & 0 & -\left(\frac{d_{f4}}{d_0}\right)^{-b} \\ 0 & 0 & 0 & 0 & -\left(\frac{d_{f5}}{d_0}\right)^{-b} & 0 & 0 \\ 0 & 0 & 0 & 0 & 0 & 0 & 0 \\ 0 & 0 & 0 & 0 & 0 & 0 & -\left(\frac{d_{f7}}{d_0}\right)^{-b} \end{pmatrix} \tag{14}$$

$$B_{in} = \begin{pmatrix} 0 & 0 & 0 & 0 & 0 & 0 & 0 \\ 0 & 0 & 0 & 0 & 0 & 0 & 0 \\ (\frac{d_{c3}}{d_0})^{-b} & 0 & 0 & 0 & 0 & 0 & 0 \\ (\frac{d_{c4}}{d_0})^{-b} & (\frac{d_{c4}}{d_0})^{-b} & 0 & 0 & 0 & 0 & (\frac{d_{c4}}{d_0})^{-b} \\ 0 & 0 & 0 & 0 & 0 & 0 & 0 \\ (\frac{d_{c6}}{d_0})^{-b} & (\frac{d_{c6}}{d_0})^{-b} & 0 & 0 & (\frac{d_{c6}}{d_0})^{-b} & 0 & (\frac{d_{c6}}{d_0})^{-b} \\ 0 & 0 & 0 & 0 & 0 & 0 & 0 \end{pmatrix} \tag{15}$$

The value of the parameter $Z$ in each surface box is specified to vary as a fraction of the global value for $Z$ in SCP-M (presently 5.0 mol C m$^{-2}$ yr$^{-1}$). There are higher fractions in the northern and southern oceans, and smaller fractions in the low latitude and polar oceans (e.g. Sarmiento and Gruber, 2006). During the model set-up we manually tuned the individual surface box values, by multiplying the global value for $Z$ by a scalar unique to each box. The values were tuned to align the model's output with GLODAPv2 data for DIC, phosphorous, alkalinity, carbonate ion and the carbon isotopes in each of the ocean boxes (Table 1). The resulting range for $Z$ (1.1-5.33 mol C m$^{-2}$ yr$^{-1}$) compares with the observations-based range of Martin et al. (1987), of 1.2-7.1 mol C m$^{-2}$ yr$^{-1}$, and Sarmiento and Gruber (2006) of 0-5 mol C m$^{-2}$ yr$^{-1}$. We chose a value for the dimensionless $b$ depth decay parameter, of 0.75, which falls in the range of Gloege et al. (2017), of 0.68-1.13, and the error range of Berelson (2001), of $0.82 \pm 0.16$. We found a global value of 0.75 to produce a better fit to the GLODAPv2 data when calibrating the model. The biological flux of other elements and species such as phosphorous and alkalinity, are calculated from the biological carbon flux using so-called "Redfield ratios" (e.g. Redfield et al., 1963; Takahashi et al., 1985; Anderson and Sarmiento, 1994).

## 2.3 pCO$_2$ and carbonate

Modelling of air-sea gas exchange, atmospheric pCO$_2$ and the "carbonate pump", rests on a realistic estimation of pCO$_2$ in the ocean. For example, only a fraction of DIC in seawater can exchange with the atmosphere, and this fraction is estimated by the oceanic pCO$_2$. DIC itself consists of three major constituents: carbonic acid, bicarbonate and carbonate. Their relative proportions depend on total DIC, alkalinity, pH, temperature and salinity (Zeebe and Wolf-Gladrow, 2001).

pCO$_2$ is estimated by subtracting alkalinity from DIC. However, this is only accurate to $\pm 10$ per cent (Sarmiento and Gruber, 2006), which may cause problems for scenario analysis and sensitivity testing within such a large error band. More complex calculations can require numerous iterations and can be computationally expensive (e.g. Toggweiler and Sarmiento, 1985; Zeebe and Wolf-Gladrow, 2001; Follows et al., 2006). We apply the routine of Follows et al. (2006) in SCP-M, which is a direct solution, rather than an iterative approach to solve for pCO$_2$ at each time step of a model run. This was demonstrated by Follows et al. (2006) to be sufficiently accurate for modelling purposes. The calculation takes inputs of DIC, alkalinity, temperature, salinity, phosphorous and silicate to estimate pCO$_2$.

**Table 1.** Values for the $Z$ biological production parameter (at 100m ocean depth) used in the SCP-M model calibration. A global value for $Z$ was tuned in each surface box using scalars (column 3) to yield unique values for each surface box (column 4).

| Model surface box | Global value at 100m ocean depth (mol C $m^{-2}$ $yr^{-1}$) | Scalar (tuned) | Model input (tuned) mol C $m^{-2}$ $yr^{-1}$ |
|---|---|---|---|
| Box 1 (Low latitude) | 5.0 | 0.22 | 1.1 |
| Box 2 (Northern) | 5.0 | 0.90 | 4.5 |
| Box 5 (Polar) | 5.0 | 0.35 | 1.75 |
| Box 7 (Subpolar) | 5.0 | 1.07 | 5.33 |

Solving for pCO$_2$ enables the calculation of the concentrations of the three species of DIC, which further enables estimation of the dissolution and burial of carbonate in the water column and sediments. The latter is an important part of the oceanic carbon and alkalinity cycles and provides important feedbacks to atmospheric CO$_2$ on thousand year timeframes (e.g. Farrell and Prell, 1989; Anderson et al., 2007; Mekik et al., 2012; Yu et al., 2014b).

### 2.3.1 The carbonate pump

According to Emerson and Hedges (2003), $\sim$20-30 per cent of CaCO$_3$ formed in the ocean's surface is preserved in ocean floor sediments, with the rest dissolved in the water column. Klaas and Archer (2002) estimated that 80 per cent of the organic matter fluxes in the ocean below 2,000m are driven by organic matter associated with carbonate ballast. Therefore, the so-called "carbonate pump" is a relatively efficient transport of carbon and alkalinity in the ocean. According to Farrell and Prell (1989), the carbonate pump is a dynamic process, and the dissolution and burial in sediments of CaCO$_3$ is observed to vary across (and within) glacial/interglacial cycles.

To replicate this flux of carbon and alkalinity, a term is added to the carbon cycle equation to represent the flux of calcium carbonate (shells) out of the surface boxes into the abyssal box and sediments. This is an extension of the surface organic carbon flux $Z$ described in Eq. (13), via the "rain ratio" parameter. The rain ratio is a common term in ocean biogeochemistry (e.g. Archer and Maier-Reimer, 1994; Ridgewell, 2003) and refers to the ratio between shell-based 'hard' carbon and organic 'soft' carbon fluxes in the biologically-driven rain of carbon from the ocean's surface. Sarmiento et al. (2002) estimated a global average value for the rain ratio of 0.06 $\pm$0.03, with local maxima and minima of 0.10 and 0.02, respectively, providing a narrow range of global values. We apply the rain ratio as a parameter multiplied by the organic flux parameter $Z$. We chose an initial value of 0.07, which provided appropriate values for DIC, alkalinity and dissolution in the model's boxes during the model spin-up (with reference to GLODAPv2 data). The combination delivers the physical production and export of calcium carbonate at the Martin reference depth (100m).

Once the production and export flux at the Martin reference depth is established, the distribution of calcium carbonate in the boxes below is a function of dissolution. According to Milliman et al. (1999), the theory that calcium carbonate only dissolves

at great depths in carbonate-undersaturated water is "*one of the oldest and most strongly held paradigms in oceanography*" (e.g. Sverdrup et al., 1941). However, in nature, the alkalinity and carbonate ion concentration profiles suggest that 30-60% of carbonate produced is dissolved in shallower water that is saturated (Harrison et al., 1993; Milliman et al., 1999). Theories for this outcome include, the emergence of locally undersaturated waters due to remineralisation of biological carbon (Jansen

et al., 2002), or, dissolution by zooplankton grazing (Milliman et al., 1999). Battaglia et al. (2016) found similar skill in model results for replicating observed dissolution profiles, whether a non- or saturation-dependent dissolution constant was used. Battaglia et al. (2016) recommended the use of a basic non-saturation-dependent (i.e. constant) dissolution parameter in Earth carbon system models for computing efficiency, with limited loss of accuracy. As such, we include two parts to the dissolution equation, a non-saturation-dependent dissolution constant, to reflect the 'unknown' processes that likely cause the observed

dissolution of calcium carbonate in waters that are saturated, and a saturation state-dependent component, using the dissolution function of Morse and Berner (1972). We include the latter to enable dynamic feedback to take place in the carbonate system after model perturbations. The saturation-dependent dissolution is a function of the average carbonate ion composition for each box, relative to its temperature and pressure-dependent saturation concentration (Morse and Berner, 1972; Millero, 1983). We choose the median depth of each box for the calculation in the ocean boxes, and the floor of the abyssal box for the sediment

surface dissolution. We assume 100% of calcium carbonate takes the form of calcite. If the surface export flux of $CaCO_3$ is greater than dissolution in the ocean boxes, then the remainder escapes to the sediments. This is a flux out of the ocean of alkalinity and carbon in the ratio of 2:1 assumed for carbonate shells (Sarmiento and Gruber, 2006). DIC and alkalinity can return to the abyssal box from the sediments via undersaturation-driven dissolution in the abyssal water overlying the sediments.

The net flux of carbonate, between ocean boxes and out of the ocean and into the sediments, is shown in vectorised Eq. (16):

$$\left[\frac{d\mathbf{C}}{dt}\right]_{carb} = \frac{(F_{CA}\mathbf{ZS})}{\mathbf{V}} + (\zeta + \epsilon)\mathbf{CaCO_3} \tag{16}$$

where $F_{CA}$ is the rain ratio, $\zeta$ is the constant background dissolution rate, $\epsilon$ is the saturation state-dependent dissolution function of Morse and Berner (1972) and Millero (1983) and $\mathbf{CaCO_3}$ is the concentration of calcium carbonate in each box.

The dissolution equation of Morse and Berner (1972) operates on $CaCO_3$, which is calculated by multiplying Ca by $CO_3^{2-}$, where Ca is estimated from salinity in each box as per Sarmiento and Gruber (2006).

### 2.3.2   Air-sea gas exchange

$CO_2$ is transported across the air-sea interface by gaseous exchange. According to Henry's Law, the partial pressure of a gas [P] above a liquid in thermodynamic equilibrium will be directly proportional to the concentration of the gas in the liquid:

$[P] = K_H C$             (17)

where $K_H$ is the solubility of a gas in mmol m$^{-3}$ atm$^{-1}$ and C is its concentration in the liquid. Many ocean models specify the air-sea gas exchange of $CO_2$ as a function of the pCO$_2$ differential between ocean and atmosphere, a $CO_2$ solubility coefficient

(e.g. Weiss, 1974), and a so-called "piston" or gas transfer velocity, which governs the rate of gas exchange, in m s$^{-1}$ (e.g. Toggweiler, 1999; Zeebe, 2012; Hain et al., 2010; Watson et al., 2015). We adopt the same approach in estimating the exchange of $CO_2$ between a surface box and the atmosphere:

$$\left[\frac{dC_1}{dt}\right]_{gas} = P_1 S_1 K_{0_1}(pCO_{2_{at}} - pCO_{2_1})\rho \tag{18}$$

where $P_1$ is the piston velocity parameter in box 1 in m s$^{-1}$. $P$ is allowed to vary in each surface box, enabling scenario analysis such as the variation of sea-ice cover in the polar box (e.g. Stephens and Keeling, 2000; Ferrari et al., 2014). $K_{0_1}$ is the solubility of $CO_2$ in mol kg$^{-1}$ atm$^{-1}$ (Weiss, 1974), subsequently converted into mol m$^{-3}$ by multiplying by sea water density $\rho$. $pCO_{2_1}$ and $pCO_{2_{at}}$ are the partial pressures of $CO_2$ in the surface ocean box 1 and atmosphere, respectively, in ppm. The equation is vectorised as follows:

$$\left[\frac{d\mathbf{C}}{dt}\right]_{gas} = \mathbf{PSK_0}(pCO_{2_{at}} - \mathbf{pCO_2})\rho \tag{19}$$

where $\mathbf{P}$ = $P_i$ (i=1,7) with zero values for non-surface boxes, and $\mathbf{K}_0$ = $K_{0_i}$ (i=1,7).

## 2.4 Sea surface temperature and salinity

Ocean box temperature and salinity are forced in SCP-M, not calculated by the model. Each box has a prescribed value for temperature and salinity, which can be time-dependent. During initialisation, the model takes box-averaged values for temperature and salinity from the GLODAPv2 database. The values can be varied between model experiments, for example Holocene versus LGM reconstructions. We argue that this approach is plausible given the availability of temperature and salinity proxy/data inputs for a range of paleo (e.g. Adkins et al., 2002; Kohfeld and Chase, 2017), modern (e.g. Olsen et al., 2016) and future scenarios (e.g. IPCC, 2013a). For the modern and future scenarios (Section 3.3), we forced temperature with time series data and projections. The temperature and salinity values feed into the calculations for ocean $pCO_2$, which further enable calculation of air-sea gas exchange and the species of DIC in seawater ($H_2CO_3$, $HCO_{3-}$ and $CO_3^{2-}$).

## 2.5 Atmosphere and terrestrial carbon cycle

SCP-M incorporates the terrestrial biosphere, continental weathering and river run-off into the ocean, plus an atmospheric radiocarbon source, volcanic and industrial emissions.

$V$ is a constant, prescribed flux of volcanic emissions of $CO_2$, in SCP-M. Toggweiler (2008) estimated this volcanic flux of $CO_2$ emissions at $4.98 \times 10^{12}$ mol year$^{-1}$ using a carbon cycle model which balanced volcanic emissions with land-based weathering sinks. The weathering of carbonate and silicate rocks also creates DIC and alkalinity runoff into the rivers, which find its way into the ocean (Amiotte Suchet et al., 2003). Relative alkalinity and DIC concentrations affect ocean $pCO_2$ and carbonate ion levels, which impacts atmospheric $CO_2$ and the dissolution and burial of carbonates (Sarmiento and Gruber, 2006). We apply the approach of Toggweiler (2008) whereby silicate and carbonate weathering fluxes of DIC and alkalinity

enter only the low latitude surface ocean box (box 1):

$$\left[\frac{dC_1}{dt}\right]_{weath} = (W_{SC} + (W_{SV} + W_{CV})AtCO_2) \tag{20}$$

where $W_{SC}$ is a constant silicate weathering term set at $0.75\text{x}10^{-4}$ mol m$^{-3}$ year$^{-1}$, $W_{SV}$ is a variable rate of silicate weathering per unit of atmosphere $CO_2$ (ppm), set to 0.5 mol m$^{-3}$ atm$^{-1}$ $CO_2$ year$^{-1}$ and $W_{CV}$ is the variable rate of

carbonate weathering with respect to atmosphere $CO_2$, set at 2 mol m$^{-3}$ atm$^{-1}$ $CO_2$ year$^{-1}$ (Toggweiler, 2008).

Alkalinity is added to the ocean in the ratio of 2:1 to DIC (Toggweiler (2008). In the case of silicate rocks, weathering is also a weak sink of $CO_2$ (e.g. Toggweiler, 2008; Hogg, 2008). The atmospheric sink of $CO_2$ is calculated by multiplying Eq. (20) by the volume of the low latitude surface ocean box (box 1) and subtracting from atmospheric $CO_2$. Equation (20) is vectorised by multiplying by a vector of boxes with only a non-zero value for box 1.

The terrestrial biosphere may act as a sink of $CO_2$ during periods of biosphere growth (e.g. post glacial regrowth), via carbon fertilisation, or a source of $CO_2$ (e.g. glacial reduction) via respiration. We employ a two-part model of the terrestrial biosphere with a long-term (woody forest) and short-term (grassland) terrestrial biosphere box as described by Raupach et al. (2011) and Harman et al. (2011), and with net primary productivity (NPP) and respiration parameters controlling the balance between uptake and release of carbon. NPP is positively affected by atmospheric $CO_2$, the so-called "carbon fertilisation" effect, (e.g.

Raupach et al., 2011). Respiration is assumed proportional to the carbon stock. The biosphere also preferentially partitions the lighter carbon isotope $^{12}$C, leading to a relative enrichment in $\delta^{13}$C in the atmosphere during net uptake of $CO_2$. The change in atmospheric $CO_2$ from the terrestrial biosphere in the model is given by:

$$\left[\frac{dAtCO_2}{dt}\right]_{NPP} = -N_{pre}RP[1 + \beta LN(\frac{AtCO_2}{AtCO_{2pre}})] + \frac{C_{stock1}}{k_1} + D_{forest} \tag{21}$$

Where N$_{pre}$ is NPP at a reference level ("pre") of atmospheric $CO_2$, $RP$ is the parameter to split NPP between the short-

term terrestrial biosphere carbon stock (fast respiration) and the long-term stock (slow respiration) (Raupach et al., 2011). $\beta$ is the parameterisation of carbon fertilisation, which causes NPP to increase (decrease) logarithmically with rising (falling) atmospheric $CO_2$ levels, and has a typical value of 0.4-0.8 (Harman et al., 2011). C$_{stock1}$ is the short-term carbon stock and k$_1$ is the timeframe for respiration in the short-term carbon stock (in years). For the long-term terrestrial biosphere, we substitute $(1 - RP)$ in place of $RP$ and C$_{stock2}$ and k$_2$ for the long-term carbon stock and respiration rate, respectively. $D_{forest}$ is a

prescribed flux of deforestation emissions, which can be switched on or off in SCP-M. A $\delta^{13}$C fractionation factor is applied to the terrestrial biosphere fluxes of carbon, effecting an increase in atmospheric $\delta^{13}$C from biosphere growth, and a decrease from respiration.

## 2.6 The complete carbon cycle equations

Equation (22) shows the full vectorised model equation for the calculation of the evolution of carbon concentration in the ocean boxes, incorporating Eq. (1-21).

$$\frac{d(\mathbf{C})}{dt} = \left[\frac{d\mathbf{C}}{dt}\right]_{phys} + \left[\frac{d\mathbf{C}}{dt}\right]_{bio} + \left[\frac{d\mathbf{C}}{dt}\right]_{carb} + \left[\frac{d\mathbf{C}}{dt}\right]_{gas} + \left[\frac{d\mathbf{C}}{dt}\right]_{weath} \tag{22}$$

5 The calculation of atmospheric $CO_2$ is:

$$\frac{dAtCO_2}{dt} = \left[\frac{dAtCO_2}{dt}\right]_{gas} + \left[\frac{dAtCO_2}{dt}\right]_{NPP} + \left[\frac{dAtCO_2}{dt}\right]_{volcs} + \left[\frac{dAtCO_2}{dt}\right]_{weath} + \left[\frac{dAtCO_2}{dt}\right]_{anth} \tag{23}$$

where the additional term $\left[\frac{dAtCO_2}{dt}\right]_{anth}$ consists of a prescribed flux of $\delta^{13}$C-depleted and $^{14}$C-dead $CO_2$ to the atmosphere from human industrial emissions, which is activated by a model switch in SCP-M.

## 2.7 Treatment of carbon isotopes

10 Carbon isotopes are an important component in SCP-M because they are key sources of proxy data. Carbon isotope fluxes are treated largely the same as DIC in SCP-M, with minor modification. For example, carbon isotopes are typically reported in delta notation ($\delta^{13}$C and $\Delta^{14}$C), which is the ‰ deviation from a standard reference value in nature. SCP-M operates with a metric mol m$^{-3}$ for the other ocean element concentrations and flux parameters. In order to incorporate $\delta^{13}$C and $\Delta^{14}$C into this metric for the operation of model fluxes, the method of Craig (1969) is applied to convert starting data values of $\delta^{13}$C and 15 $\Delta^{14}$C from delta notation in ‰, into mol m$^{-3}$:

$$^{13}C_i = (\frac{\delta 13C_i}{1000} + 1)RC_i \tag{24}$$

Where $^{13}C_i$ is the $^{13}$C concentration in box i in mol m$^{-3}$, $\delta^{13}$C$_i$ is $\delta^{13}$C in ‰ in box i, $R$ is the $\frac{^{13}C}{^{12}C}$ ratio of the standard (0.0112372 as per the Pee Dee Belemnite value) and $C_i$ is the DIC concentration C in box i, in mol m$^{-3}$.

The calculation in Eq. (24) derives the fraction $\frac{^{13}C}{^{12}C}$ for the data or a model starting value, multiplies that value by the standard 20 reference value and then by the starting model concentration for DIC ($C_i$) in each box. This approach rests on the assumption that the fraction $\frac{^{13}C}{^{12}C}$ is the same as $\frac{^{13}C}{\text{total carbon}}$ . For example, there are three isotopes of carbon, each with different atomic weights. They occur in roughly the following abundances: $^{12}$C $\sim$98.89%, $^{13}$C $\sim$1.11% and $^{14}$C $\sim$1x10$^{-10}$%. Therefore, the assumption that $\frac{^{13}C}{^{12}C} = \frac{^{13}C}{\text{total carbon}}$, is a valid approximation. Once converted from $\delta^{13}$C (‰) to $^{13}$C in mol m$^{-3}$, SCP-M's ocean parameters can operate on $^{13}$C concentrations in each box, according to the same model flux equations used for DIC and $CO_2$. 25 The $^{13}$C model results are then converted back into $\delta^{13}$C notation at the end of the model run, in order to compare the model output with the data, which is reported in $\delta^{13}$C format. The same method is applied to $\Delta^{14}$C. The reference standard value for $\frac{^{14}C}{^{12}C}$ is 1.2x10$^{-12}$ (Craig, 1969). Where fractionation of carbon isotopes takes place, such as biological or air-sea gas exchange, fractionation factors are simply added to the model flux equations.

### 2.7.1 Biological fractionation of carbon isotopes

Biological processes influence the carbon isotopic composition of the ocean (e.g. Fontugne and Duplessy, 1978). When photosynthetic organisms are active in shallow ocean waters, they preferentially partition $^{12}$C (the lighter carbon isotope). This activity enriches the surface ocean in $^{13}$C, and relatively enriches the underlying waters in $^{12}$C when remineralisation occurs. As such, the ocean displays depletion in $\delta^{13}$C in the deep ocean relative to the shallow ocean (e.g. Curry and Oppo, 2005). In SCP-M, a fractionation factor, $f$, is simply multiplied by the biological flux in Eq. (13), to calculate marine biological fractionation of $^{13}$C and to replicate ocean distributions of $\delta^{13}$C:

$$\left[\frac{d^{13}C_i}{dt}\right]_{^{13}bio} = f * S_{st} \tag{25}$$

Where $f$ is the biological fractionation factor for stable carbon (e.g. $\sim$0.977 in Toggweiler and Sarmiento (1985)), and $S_{st}$ is the ratio of $^{13}$C to $^{12}$C in the reference standard. The typical $\delta^{13}$C composition of marine organisms is in the range -23 to -30‰. The same method is applied for biological fractionation of $^{14}$C, but with a different fractionation factor (Toggweiler and Sarmiento, 1985).

### 2.7.2 Fractionation of carbon isotopes during air-sea gas exchange

Fractionation of carbon isotopes takes place during air-sea exchange (e.g. Mook et al., 1974; Siegenthaler and Munnich, 1981; Lynch-Stieglitz et al., 1995). The lighter isotope, $^{12}$C, preferentially partitions into the atmosphere as a net effect of bidirectional gaseous exchange. This fractionation leads to the heavily depleted $\delta^{13}$C signature for the atmosphere, relative to the ocean. The approach to capture this effect in SCP-M is per Siegenthaler and Munnich (1981):

$$\left[\frac{d^{13}C_i}{dt}\right]_{^{13}gas} = \lambda[\tau R_{At} pCO_{2At} - \pi R_i pCO_{2i}] \tag{26}$$

Where $\lambda$, the "kinetic fractionation effect" (Zhang et al., 1995), accounts for the slower equilibration rate of carbon isotopes $^{13}$C and $^{14}$C across the air-sea interface, compared with $^{12}$C (Zhang et al., 1995). $R_{At}$ is the ratio of $^{13}$C to $^{12}$C in the atmosphere, $R_i$ is the ratio of $^{13}$C to $^{12}$C in surface ocean box $i$. pCO$_{2At}$ is the atmospheric pCO$_2$ and pCO$_{2i}$ is the pCO$_2$ in the surface ocean boxes. $\tau$ and $\pi$ are the fractionation factors of carbon isotope from air to sea and sea to air, respectively. These are temperature-dependent reactions and are calculated in SCP-M using the method of Mook et al. (1974).

### 2.7.3 Source and decay of radiocarbon

Natural radiocarbon is produced in the atmosphere from the collision of cosmic ray-produced neutrons with nitrogen. The production rate is variable over time and can be influenced by changes in solar winds and the earth's geomagnetic field intensity (Key, 2001). A mean production rate of 1.57 atom m$^{-2}$ s$^{-1}$ was estimated from the long term record preserved in tree-rings, although more recent estimates approach 2 atom m$^{-2}$ s$^{-1}$(Key, 2001). For use in SCP-M, this estimate needs to be converted into mols s$^{-1}$. We first convert atoms to mols by dividing through by Avogrado's number ($\sim$6.022x10$^{23}$). The resultant figure is

**Table 2.** Ocean and atmosphere data sources for the SCP-M modern carbon cycle calibration, projections and LGM-Holocene experiment.

| Indicator | Reference |
|---|---|
| Atmosphere $CO_2$ | Marcott et al. (2014), Scripps $CO_2$ Program |
| Atmosphere $\delta^{13}C$ | Schmitt et al. (2012), Scripps $CO_2$ Program |
| Atmosphere $\Delta^{14}C$ | Nydal and Lövseth (1996), Stuiver et al. (1998), Reimer et al. (2009), Turnbull et al. (2016) |
| Ocean $\delta^{13}C$ | Peterson et al. (2014) |
| Ocean $\Delta^{14}C$ | Skinner and Shackleton (2004); Marchitto et al. (2007); Barker et al. (2010); Bryan et al. (2010); Skinner et al. (2010); Burke and Robinson (2012); Davies-Walczak et al. (2014); Skinner et al. (2015); Chen et al. (2015); Hines et al. (2015); Sikes et al. (2016), Ronge et al. (2016), Skinner et al. (2017) |
| Ocean carbonate ion | Yu et al. (2014b), Yu et al. (2014a) |
| Modern ocean data (e.g. DIC, alkalinity, phosphorus, $\delta^{13}C$, $\Delta^{14}C$) | GLODAPv2 (Olsen et al., 2016) |
| Suess and bomb radiocarbon effect corrections | Broecker et al. (1980), Key (2001), Sabine et al. (2004), Eide et al. (2017) |

The late Holocene is chosen as the initial model calibration due to the absence of industrial-era $CO_2$ and bomb radiocarbon. Scripps $CO_2$ Program data originally sourced from http://scrippsco2.ucsd.edu, data currently being transitioned to http://cdiac.ess-dive.lbl.gov. The Peterson et al. (2014) database incorporates ∼500 core records across the LGM and late Holocene periods.

multiplied by the earth's surface area ($\sim5.1 \times 10^{18}$ cm$^{-2}$) to yield a production rate of $1.3296 \times 10^{-5}$ mols s$^{-1}$. This source rate, divided through by the molar volume of the atmosphere, is added to the equation for atmospheric radiocarbon concentration. A decay timescale for radiocarbon of 8,267 years, is applied to each box in the model.

## 3 Modelling results

5    The modern carbon cycle has been extensively modelled as part of efforts to understand the impact of anthropogenic emissions on climate. There is abundant data on emissions and detailed observations of the modern carbon cycle with globally coordinated ocean surveys and land-based measuring stations. In addition, numerous modelling exercises, using consensus-type emissions projection scenarios from the Intergovernmental Panel on Climate Change (IPCC), have created a body of modelling inputs and results. This provides an ideal testing ground for SCP-M. We first calibrate the model for the preindustrial period, then
10   simulate historical and projected human emissions under a number of scenarios.

## 3.1 Preindustrial calibration

We choose the late Holocene period (6-0.2 ka) for our calibration because it has relatively good proxy data coverage (e.g. Table 2) and a relatively steady climate in the absence of perturbations such as industrial $CO_2$ emissions, bomb radiocarbon or glacial terminations. The late Holocene is also close to the preindustrial period (1700's) in order to act as a starting point for modern
carbon cycle simulations. To calibrate the model for the late Holocene we begin with the modern day GLODAPv2 dataset (https://www.nodc.noaa.gov/ocads/oceans/GLODAPv2/) which we average into the model's boxes on depth and latitude co-ordinates, using one of the SCP-M scripts (Fig. 2). The GLODAPv2 database incorporates data from ∼1 million seawater samples from 700 cruises over the years 1972-2013, including data from the original GLODAP dataset, plus CARINA and PACIFICA datasets (Olsen et al., 2016). We assume an average data year of 1990 for the data accumulated over the period
1972-2013. We make adjustments to the ocean concentrations of DIC, $\delta^{13}C$ and $\Delta^{14}C$ for the effects of industrial emissions (the "Suess" effect) and bomb radiocarbon in the atmosphere using published estimates (Broecker et al., 1980; Key, 2001; Sabine et al., 2004; Eide et al., 2017). For example, Eide et al. (2017) establishes a mathematical relationship between Suess $\delta^{13}C$ and CFC-12 in the ocean, which we applied using GLODAPv2 CFC-12 data to correct the ocean $\delta^{13}C$ data. We force the model with late Holocene average data for atmosphere $CO_2$, $\delta^{13}C$ and $\Delta^{14}C$ (data sources in Table 2). The model's starting
parameters are set from literature values (Table 6, Appendix), including the point estimates for ocean circulation and mixing fluxes from Talley (2013). Using the Suess- and bomb- adjusted GLODAPv2 ocean dataset, and late Holocene atmosphere data, as the starting point, combined with the literature-determined parameter values, the model is allowed to run freely for 15 kyr in spin-up. This spin-up is ample time for model equilibration and to allow slower processes such as carbonate compensation to take effect. The resulting model equilibrium ocean and atmosphere element concentrations from the spin-up are stored
and then carried forward as the starting data for subsequent late Holocene simulations. Figure 3 shows the results of the model spin up (red stars), compared with late Holocene atmosphere data and their standard error (blue dots and error bars) across the time period. We also show the model results compared with late Holocene ocean data from various sources (Table 2) which is averaged into the box model regions for comparison.

The late Holocene calibration convincingly satisfies the atmospheric data values for $CO_2$, $\delta^{13}C$ and $\Delta^{14}C$. Model results are
also in good agreement with the late Holocene ocean $\Delta^{14}C$, falling within error or very close for all boxes covered by data. The surface boxes (1, 2) are relatively enriched in $\Delta^{14}C$ relative to deeper boxes, reflecting their proximity to the atmospheric source of $^{14}C$, although the spread of values across the ocean boxes is narrow. The surface boxes (1, 2 and 7) intuitively display more enriched $\delta^{13}C$ than the intermediate (3), deep (4) and abyssal (6) boxes, mainly due to the effects of the biological pump. For most of the model's boxes, the results fall within the standard error of the late Holocene $\delta^{13}C$ data. The Southern Ocean box
(5), is an exception due to its extensive vertical coverage of 2,500m incorporating the surface boundary with the atmosphere and the deep ocean, coupled with the sparse $\delta^{13}C$ core data for the polar Southern Ocean (one data point, no error bars). SCP-M also exaggerates the depletion in $\delta^{13}C$ in the deep box (4), relative to the data observation.

There is limited data coverage for carbonate ion proxy ($CO_3^{2-}$), although the model replicates the available data well. $CO_3^{2-}$ concentrations can be interpreted as alkalinity less DIC (Zeebe and Wolf-Gladrow, 2001; Yu et al., 2014b), for the purposes

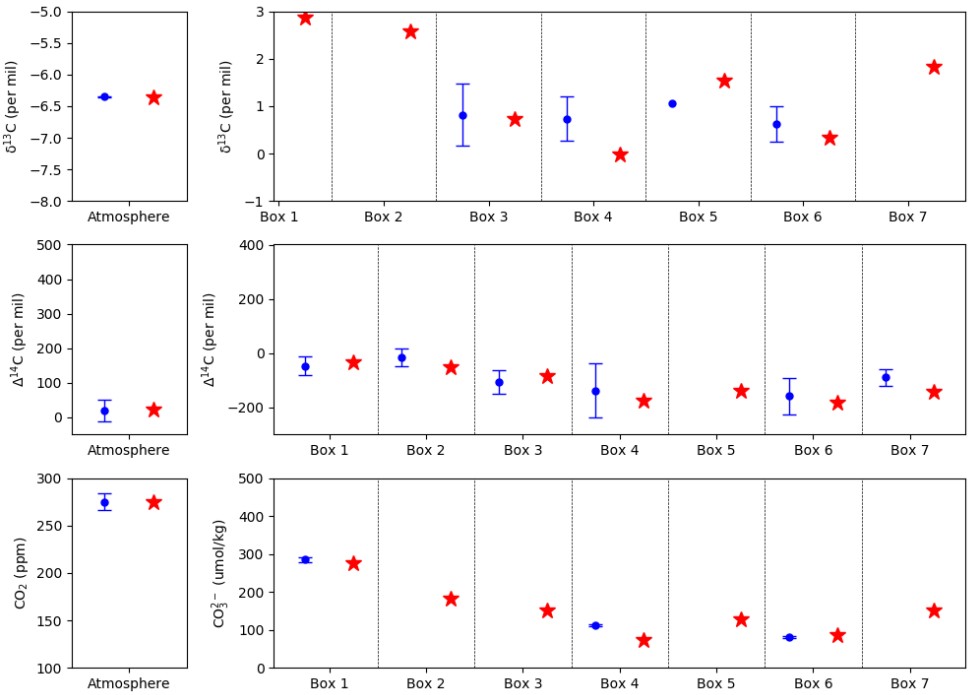

**Figure 3.** SCP-M late Holocene-calibrated model results using model input parameters from the literature (Table 6). Left panels show model results for atmospheric $\delta^{13}$C, $\Delta^{14}$C and $CO_2$ (red stars) plotted against late Holocene average data values (blue dots) with standard error bars. The right panel shows the model results for oceanic $\delta^{13}$C, $\Delta^{14}$C and carbonate ion proxy (red stars) plotted against late Holocene average ocean data where available (blue dots). Data sources are shown in Table 2.

of analysing model results charts. $CO_3^{2-}$ is relatively abundant in the surface boxes (e.g. boxes 1 and 2), reflecting the higher amount of alkalinity relative to carbon due to the soft tissue biological pump, which prioritises organic carbon export over alkalinity export. $CO_3^{2-}$ is less abundant in the deep ocean (boxes 4 and 6), because there is more carbon relative to alkalinity due to remineralisation of organic matter, a pattern that SCP-M replicates.

## 3.2 Sensitivity tests

We undertook parameter sensitivity tests to understand changes in atmospheric $CO_2$, $\Delta^{14}$C and $\delta^{13}$C in SCP-M. This serves two purposes: 1) to understand the directional relationship between the parameter values and these key model outputs, and evaluate whether they make sense, and 2) to inform the LGM-Holocene model-data experiment in the following section. For example, if the GOC parameter $\Psi_1$ displays a negative relationship with atmospheric $CO_2$, it would make sense to probe parameter values lower than modern, to replicate the lower atmospheric $CO_2$ in the LGM. We varied parameter values around their modern day settings in 10 kyr model runs, and plotted the output against atmospheric $CO_2$, $\Delta^{14}$C and $\delta^{13}$C (Fig. 4). For

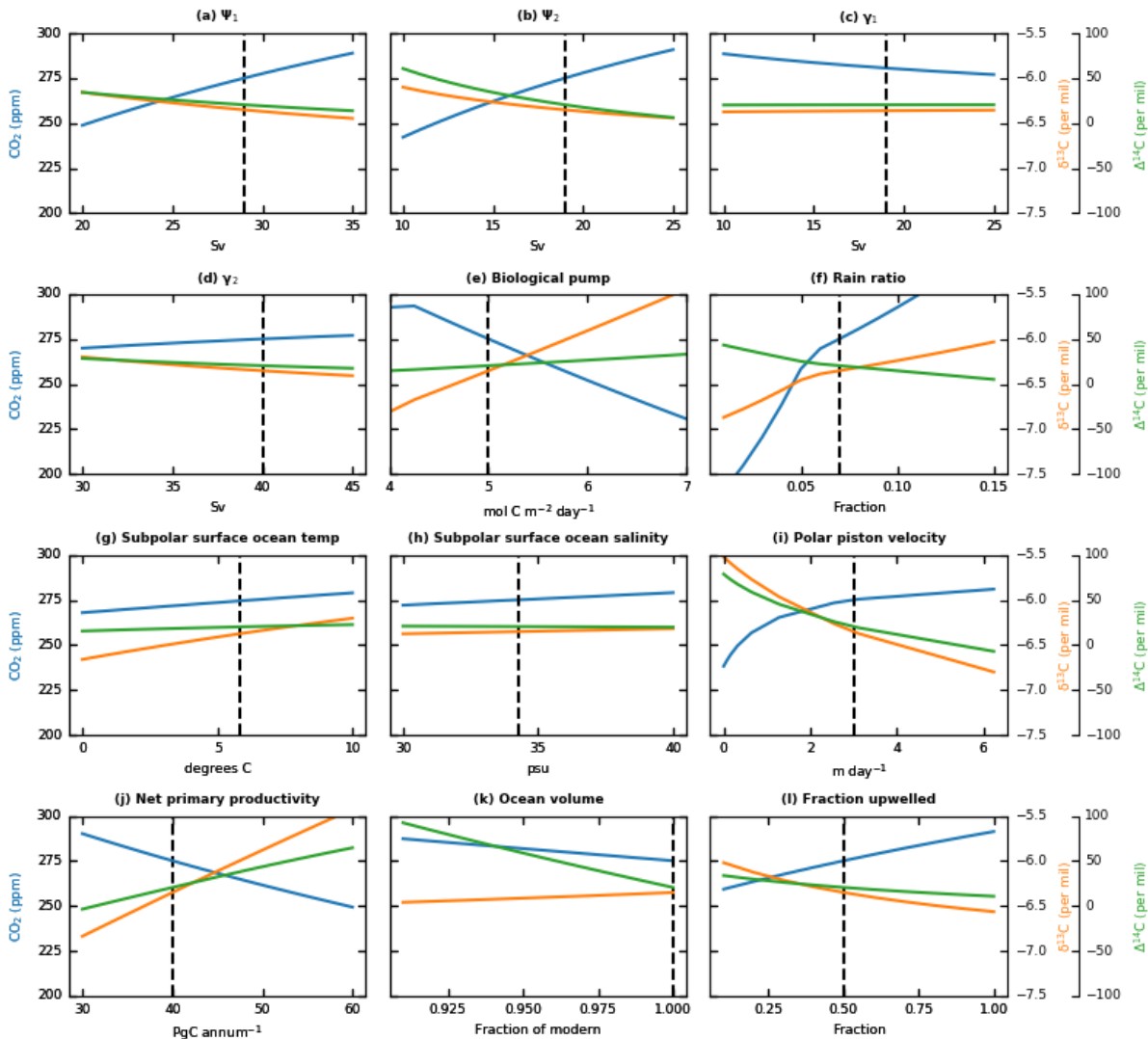

**Figure 4.** Univariate parameter sensitivity tests around modern day estimated values, plotted for atmospheric $CO_2$, $\Delta^{14}C$ and $\delta^{13}C$ versus (a) $\Psi_1$, (b) $\Psi_2$, (c) $\gamma_1$, (d) $\gamma_2$, (e) biological pump, (f) rain ratio, (g) subpolar surface box temperature, (h) subpolar surface box salinity, (i) polar box piston velocity, (j) net primary productivity, (k) ocean volume and (l) fraction of deep water upwelled into the subpolar surface box. We varied parameter input values as plotted on the x-axes and show model output for atmospheric $CO_2$, $\Delta^{14}C$ and $\delta^{13}C$. Atmospheric $CO_2$ show the greatest sensitivity to parameters associated with ocean circulation, biology and the terrestrial biosphere. Other parameters exert less influence on atmospheric $CO_2$ but are important for atmospheric carbon isotope values. Modern day estimates used in SCP-M are shown with vertical black dotted lines in each subplot (sources in the text and Appendix Table 6).

example, Fig. 4(a-d) shows sensitivity variations above and below the model's modern values for ocean circulation and mixing parameters, sourced from Talley (2013) and Toggweiler (1999). Atmospheric $CO_2$ is very sensitive to $\Psi_1$ and $\Psi_2$ but displays limited response to $\gamma_1$ and $\gamma_2$ over the ranges analysed (Fig. 4(a-d)). Atmospheric $\Delta^{14}C$ and $\delta^{13}C$ are negatively related to $\Psi_1$ and $\Psi_2$. The slower ocean turnover leads to a reduced rate of upwelling and surface de-gassing of $\Delta^{14}C$- and $\delta^{13}C$-depleted waters, causing higher values in the atmosphere. The effect of the mixing parameters on the atmosphere variables is muted because they have limited impact on the upwelling regime for carbon, with any upward flux of carbon offset by a downward flux (mixing).

The soft tissue export flux parameter, $Z$, displays an inverse relationship with $CO_2$ (Fig. 4(e)). A higher global value of $Z$ drives the removal of carbon from the surface ocean, and the resulting $CO_2$ flux into the ocean from the atmosphere decreases $CO_2$. Lower $Z$ leads to increased atmospheric $CO_2$. $\delta^{13}C$ is particularly sensitive to $Z$, moving it well away from modern (and therefore Holocene and LGM) values from a minor perturbation. The rain ratio (Fig. 4(f)) increases $pCO_2$ in the surface ocean boxes, leading to de-gassing of $CO_2$ to the atmosphere, and therefore modestly decreasing atmospheric $\Delta^{14}C$, as the lighter $^{12}C$ is preferentially partitioned across the air-sea interface.

Increasing surface ocean box temperature (Fig. 4(g)) increases atmospheric $CO_2$, an intuitive outcome given that warmer water absorbs less $CO_2$ (Weiss, 1974), and SCP-M employs a temperature- and salinity- dependent $CO_2$-solubility function. Air-sea fractionation of $\delta^{13}C$ also decreases with higher temperatures, leading to higher atmospheric $\delta^{13}C$. According to Mook et al. (1974), air-to-sea fractionation of $\delta^{13}C$ (making the atmosphere more depleted in $\delta^{13}C$) increases at a rate of approximately $0.1\permil\,°C^{-1}$ of cooling. SCP-M employs temperature-dependent air-sea gas $\delta^{13}C$ fractionation factors (Mook et al., 1974). $\Delta^{14}C$ is invariant to box temperature as the fractionation parameters employed in the model are non temperature dependent. $CO_2$ displays a weak positive relationship with surface ocean box salinity (Fig. 4(h)), due to the decreasing solubility of $CO_2$ in ocean water with increasing salinity (Weiss, 1974).

As the polar box piston velocity $P$ slows down (Fig. 4(i)), atmospheric $CO_2$ falls. At lower values of $P$ the polar box, a region of outgassing of $CO_2$ due to the upwelling of deep-sourced carbon-rich water in that part of the ocean, will exchange $CO_2$ with the atmosphere at a slower rate. The reduced outgassing of $\delta^{13}C$-depleted carbon to the atmosphere with a lower $P$, leads to higher $\delta^{13}C$ values in the atmosphere. Atmospheric $\Delta^{14}C$ increases with a slowing of $P$ as the pathways for it to invade the ocean from its atmospheric source, are slower, and there is reduced outgassing of old, low $\Delta^{14}C$ waters.

Terrestrial biosphere NPP (Fig. 4(j)) is a sink of $CO_2$ and fractionates the ratios of the isotopes of carbon, leading to higher values for $\delta^{13}C$ and to a lesser extent, $\Delta^{14}C$, in the atmosphere. It is likely that NPP plays a feedback role and modulates $CO_2$, $\delta^{13}C$ and $\Delta^{14}C$ (Toggweiler, 2008). Varying the ocean surface area (Fig. 4(k)) has modest impacts on $CO_2$ and $\delta^{13}C$, but a large impact on $\Delta^{14}C$. Decreasing the ocean volume leads to a lower surface area for $CO_2$ and atmospherically-produced radiocarbon to enter the ocean, causing them to increase in the atmosphere. We expect that changing the ocean surface area (from sea level), and therefore volume, leads to changes in $pCO_2$ on glacial/interglacial timescales. Increasing the fraction of deep water upwelled into the subpolar box (Fig. 4(l)), intuitively raises $CO_2$ but lowers $\delta^{13}C$ and $\Delta^{14}C$, by upwelling carbon rich and isotopically-depleted water to the ocean surface boxes.

### 3.3 Modern carbon cycle simulation

Human fossil fuel and land-use change emissions have contributed ~575 Gt carbon to the atmosphere between 1751 and 2010 (Boden et al., 2017; Houghton, 2010) and up until 2014 were growing at an accelerating rate. In response, the Earth's carbon cycle continually partitions carbon between its component reservoirs, with positive and negative feedbacks. The net effect is a build-up of carbon in most reservoirs. Given the dominance of the anthropogenic emissions source in the modern global carbon cycle, a simulation model should be able to provide a plausible replication of its effects. We modelled the effects of anthropogenic emissions and atmospheric nuclear bomb testing on the carbon reservoirs and fluxes in SCP-M. The experiment forces the late Holocene/preindustrial SCP-M equilibrium described in Section 3.1, with estimates of industrial fossil fuel and land use change $CO_2$ emissions, sea surface temperature (SST) changes and atmospheric bomb $^{14}C$ fluxes, from historical data dating from 1751. For the future years to 2100, we force the model with the IPCC's representative concentration pathway (RCPs) $CO_2$ emissions and SST scenarios (Boden et al., 2017; Houghton, 2010; IPCC, 2013a). We compare the model results with atmospheric $CO_2$, $\delta^{13}C$ and $\Delta^{14}C$ historical data, and published modelling results for future years from the CMIP5 project (https://cmip.llnl.gov/cmip5/).

Figure 5 shows the modern carbon cycle simulation using SCP-M, compared with historical atmospheric data for $CO_2$, $\delta^{13}C$ and $\Delta^{14}C$ and GLODAPv2 ocean data (estimated data year 1990). Importantly, SCP-M provides an appropriate simulation of the carbon cycle response to the human emissions inputs by replicating the atmospheric patterns for $CO_2$, $\delta^{13}C$ and $\Delta^{14}C$ preserved in data observations for the period 1751-2016 (a-b). The atmospheric $CO_2$ and $\delta^{13}C$ data is sourced from the Scripps $CO_2$ program (originally sourced from http://scrippsco2.ucsd.edu, data currently being transitioned to http://cdiac.ess-dive. lbl.gov), and $\Delta^{14}C$ data is sourced from Nydal and Lövseth (1996), Stuiver et al. (1998) and Turnbull et al. (2016). A key feature of the historical data is the substantial increase in human emissions from circa 1950 onwards which is accompanied by higher atmospheric $CO_2$ and a steep drop in $\delta^{13}C$ (Fig. 5(a)), which reflects the $\delta^{13}C$-depleted anthropogenic emissions. The effect of emissions on atmospheric $\Delta^{14}C$ (Fig. 5(b)) in the 20th century is largely overprinted by the influence of bomb radiocarbon. Emissions are seen as a slight downturn in the model and data $\Delta^{14}C$ in the immediate lead up to the release of bomb radiocarbon into the atmosphere, and the downward trend from ~2020 onwards. The spike in $\Delta^{14}C$ during the period of bomb radiocarbon release, lasts during the period 1954-1963 and then disperses as $^{14}C$ is absorbed by the ocean. The simulation shows that SCP-M agrees with the GLODAPv2 ocean data by 1990 (Fig. 5(c-f)), with most boxes falling within the standard deviation of average data values, lending confidence to the model's simulation of the redistribution of carbon.

Figure 6 shows the emissions profile (a) and modelling results for atmospheric $CO_2$ (b) over historical time and projected forward to 2100 for the IPCC RCPs. The SCP-M output falls below the IPCC projections for atmospheric $CO_2$ in RCP2.6 and 4.5, but provides a close match with RCP6.0 and 8.5.

Figure 7(a) shows the the annual uptake of $CO_2$ by the ocean, modelled with SCP-M. The model begins the period close to a steady state between the atmosphere and surface ocean $pCO_2$, with limited transfer across the interface. Beginning circa 1950 the ocean takes up an increased load of $CO_2$ from the atmosphere. By 2100, SCP-M models a range of annual $CO_2$ uptake by the ocean of 0-6 PgC annum$^{-1}$ across the RCPs. This is similar to the range of values estimated by the CMIP5 models (also

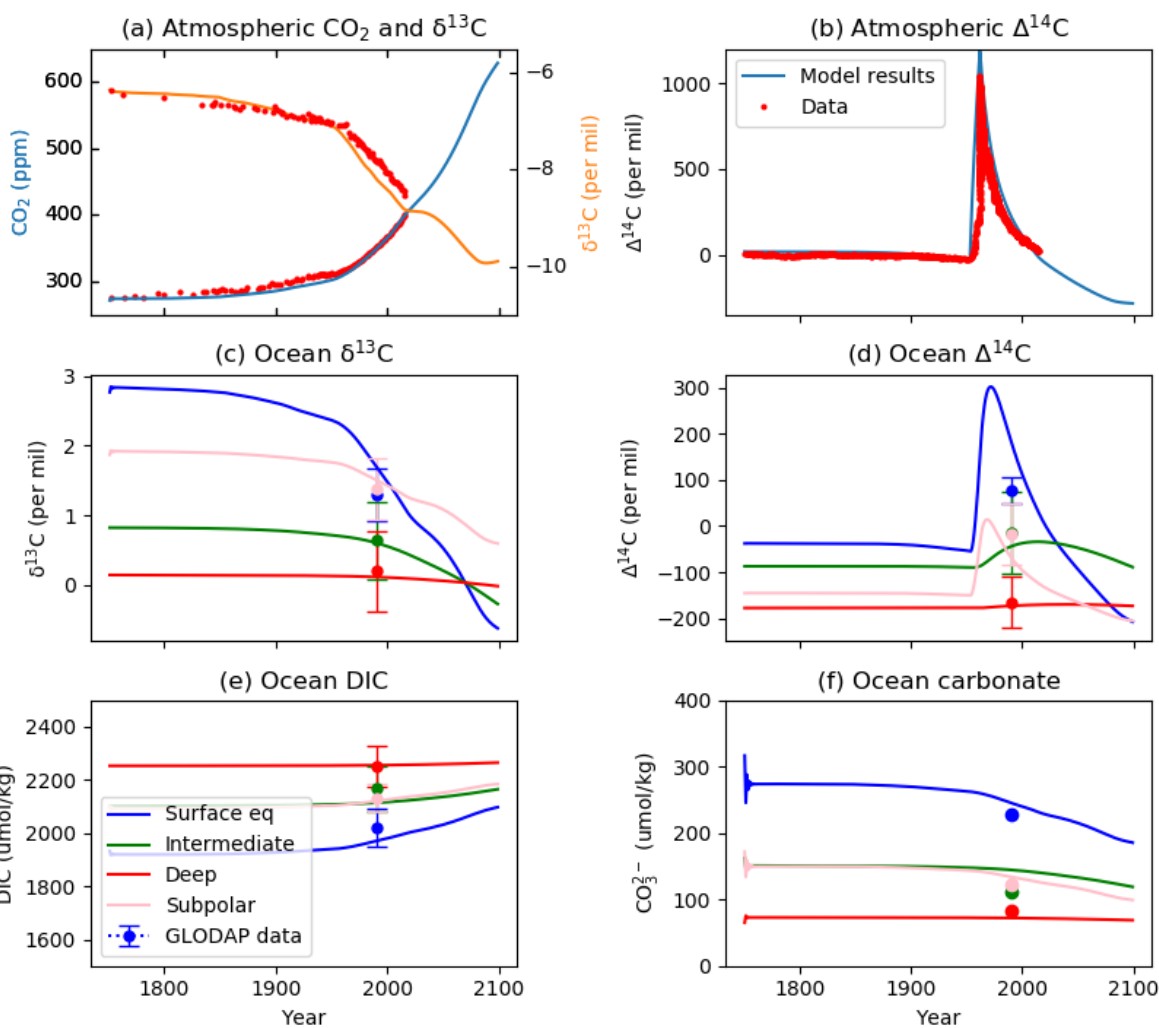

**Figure 5.** SCP-M modelling results compared with modern atmospheric and ocean GLODAPv2 data for (a) atmospheric $CO_2$ and $\delta^{13}C$, (b) atmospheric $\Delta^{14}C$, (c) ocean $\delta^{13}C$, (d) $\Delta^{14}C$, (e) DIC and (f) carbonate ion proxy. Projections beyond 2016 include the RCP6.0 emissions trajectory. In the top row we plot SCP-M model results for $CO_2$, $\delta^{13}C$ and $\Delta^{14}C$ (lines) for the period 1751-2100 against atmospheric data for $CO_2$, $\delta^{13}C$ and $\Delta^{14}C$ (red dots). The SCP-M model output closely resembles the atmospheric data record. The perturbation from industrial-era, isotopically depleted ($\delta^{13}C$) and dead ($\Delta^{14}C$) $CO_2$ is clear, as is the impact of atmospheric nuclear tests on $\Delta^{14}C$ during 1954-1963. In the other rows we plot SCP-M model results (boxes as shown) versus GLODAPv2 data (dots/error bars, same colour as corresponding boxes). We assume an average data year of 1990 for the GLODAPv2 data accumulated over the period 1972-2013. For most of the SCP-M ocean boxes, the model results fall within or very close to error ranges of the GLODAPv2 data, despite large perturbations in the model and data from industrial-era emissions and bomb radiocarbon.

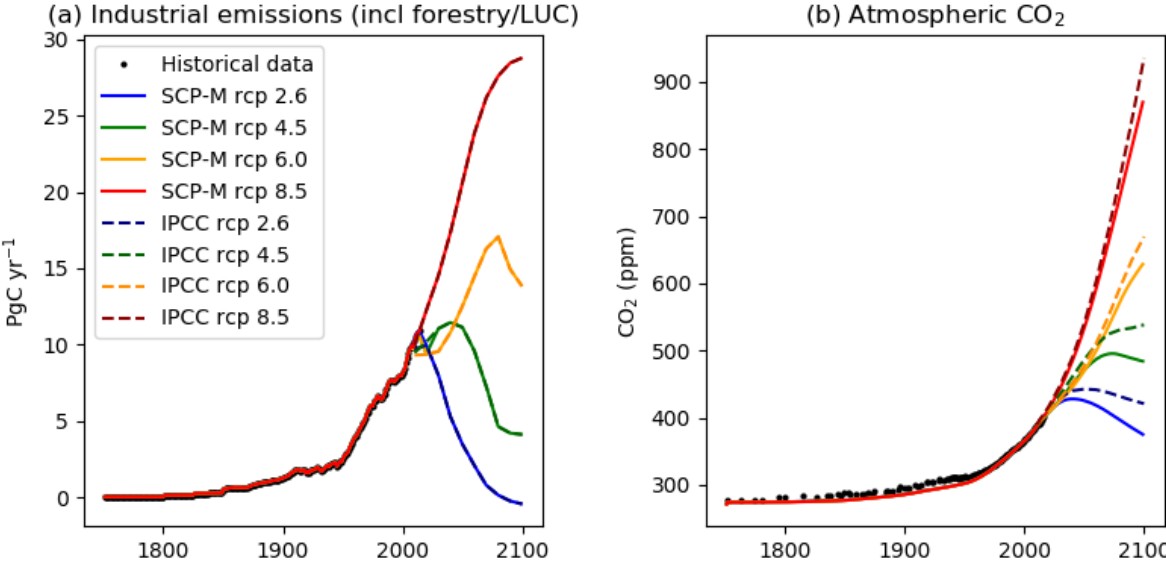

**Figure 6.** SCP-M RCP modelling results compared with IPCC emissions and $CO_2$ scenarios. Panel (a) shows the IPCC's RCP emissions pathways out to 2100 which are inputted to SCP-M for the modern carbon cycle simulation. Panel (b) shows SCP-M model output for atmospheric $CO_2$ (firm lines) plotted against IPCC atmospheric $CO_2$ projections for the RCP pathways (dashed lines). The SCP-M output undershoots the IPCC projections for RCP2.6 and 4.5, but provides a close match with RCP6.0 and 8.5.

shown in Fig. 7(a)), reproduced from Jones et al. (2013). The cumulative uptake of emissions by the ocean over the period 1751-2100 (Fig. 7(b)) modelled by SCP-M of ~350-750 PgC, is at the upper end of the modelled range of CMIP5 models of ~200-600 PgC over the period 1850-2100 (Jones et al., 2013). The SCP-M simulations commence in 1751 and therefore incorporate an extra 100 years of fossil fuel and land use change emissions beyond the CMIP5 model results presented in Jones
et al. (2013). Wang et al. (2016) quote a range of 412-649 PgC cumulative uptake by the ocean by 2100 from 11 CMIP5 models, a closer range to the SCP-M outcomes. Figure 8 shows the partitioning of anthropogenic $CO_2$ emissions into the carbon cycle reservoirs in RCP6.0 by 2100, as simulated with SCP-M and compared with modelling results presented by the IPCC for the same scenario (IPCC, 2013b). By this time, the load of human emissions is roughly 45:55 split between the atmosphere and the combined terrestrial biosphere and ocean.

By 2100, in RCP6.0 the carbon cycle is substantially changed from the preindustrial/late Holocene state. This is the result of the accumulation of hundreds of years of human industrial $CO_2$ emissions in the atmosphere and other reservoirs (Fig. 9). Anthropogenic $CO_2$ emissions transfer carbon to the atmosphere, ocean and terrestrial biosphere. The fluxes between the carbon reservoirs also change. In the preindustrial state, $CO_2$ enters the ocean in the low latitudes and northern ocean (shown as negative fluxes in Fig. 9), and de-gasses in the Southern Ocean (positive flux) under the influence of ocean upwelling in
that region. In RCP6.0, the atmospheric $CO_2$ concentration increases to the extent that the atmosphere-ocean $pCO_2$ gradient

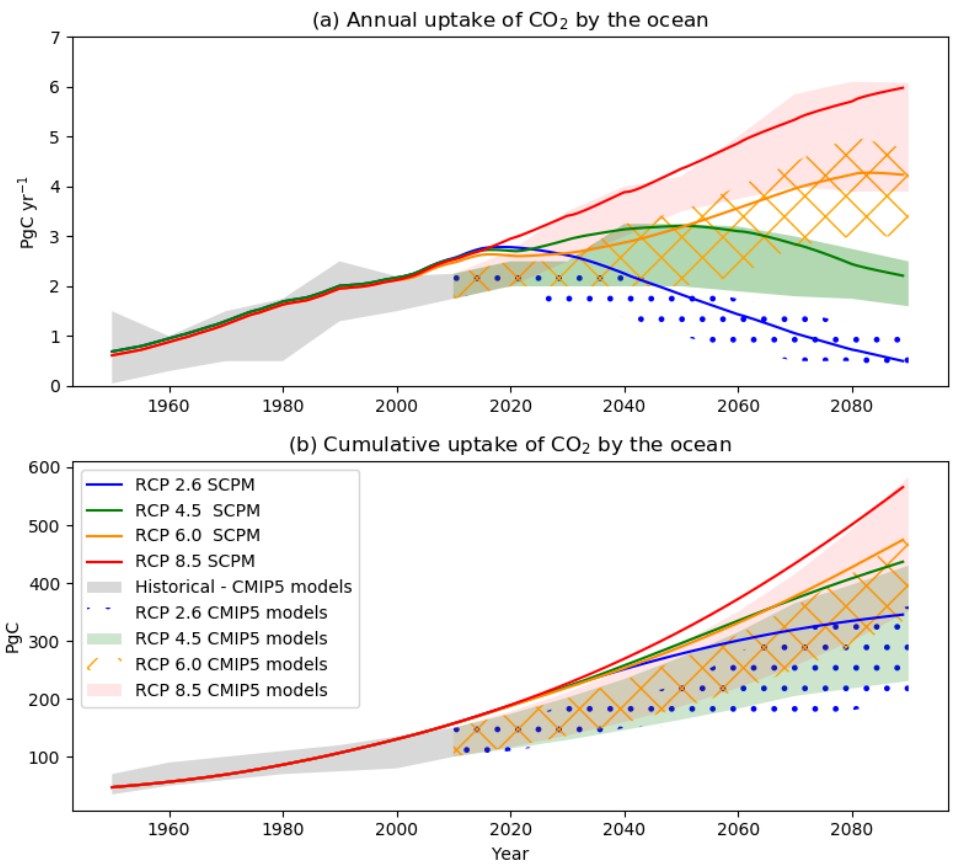

**Figure 7.** Panel (a) shows shows the annual uptake of $CO_2$ by the ocean in each of the RCPs over the period 1751-2100, modelled with SCP-M. By 2100, SCP-M estimates a range of 0-6 PgC year$^{-1}$ across the RCPs as estimated by CMIP5 models, reproduced from Jones et al. (2013). Panel (b) shows the cumulative uptake of $CO_2$ by the ocean over the same period modelled with SCP-M and compared with CMIP5 models (Jones et al., 2013).

drives all surface ocean boxes to take carbon from the atmosphere (shown as large negative changes in the air-sea fluxes of carbon, in red text in Fig. 9), despite warmer surface ocean temperatures towards the end of the projection (time series inputs). The terrestrial biosphere influx of carbon is dramatically increased by the carbon fertilisation effect, leading to a larger biomass stock which in turn also causes more respiration - both inward and outward biosphere fluxes of $CO_2$ are therefore 5 greatly enhanced. The weathering of silicate rocks on the continents, a weak sink of carbon, also accelerates under the effects of burgeoning atmospheric $CO_2$, transferring carbon from the atmosphere to the ocean via rivers. The physical fluxes of carbon within the ocean are only modestly affected, with the main exception being low latitude thermocline mixing, which in RCP6.0 mixes a larger amount of carbon back into the surface ocean box from intermediate depths. The altered balance of DIC:alkalinity, particularly in the abyssal box, leads to a decrease in the carbonate ion concentration of abyssal waters, late in

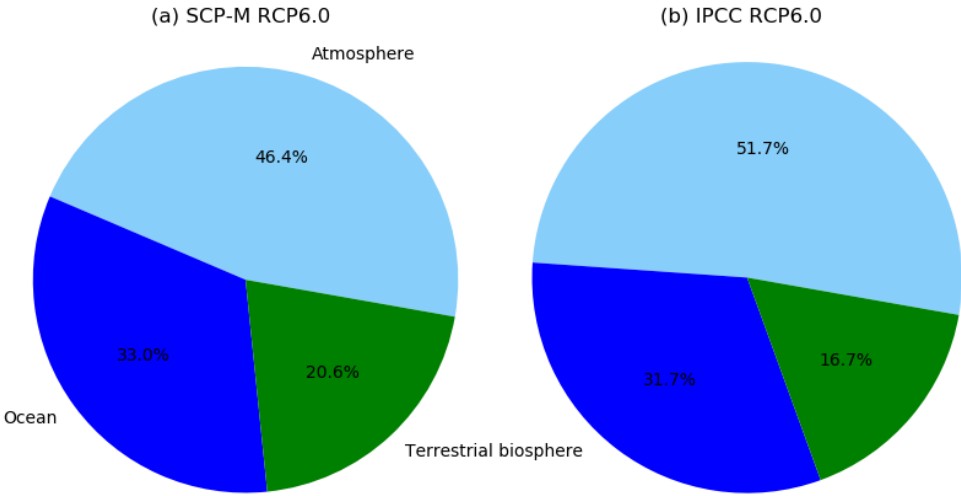

**Figure 8.** Relative uptake of $CO_2$ across the major carbon reservoirs by 2100 in RCP6.0 as modelled by SCP-M (left panel). By 2100, SCP-M projects that 46% of industrial-era emissions remain in the atmosphere, 33% reside in the ocean and 21% in the terrestrial biosphere. Shown on the right panel are results from Earth system models reproduced from the IPCC Working Group 1 5th Assessment Report, Chapter 6 (IPCC, 2013b).

the projection period, which in turn causes more dissolution of marine sediments. By 2100 this feedback brings more carbon back into the ocean, increased from 0.2 to 1.1 PgC yr$^{-1}$, but also alkalinity (in a ratio of 2:1 to DIC), thereby lowering whole of ocean pCO$_2$ - a modest negative feedback. In summary, SCP-M provides an appropriate simulation of historical atmospheric $CO_2$, $\delta^{13}C$ and $\Delta^{14}C$ data, when forced with anthropogenic $CO_2$ emissions data over the same period. For the forward-looking
RCP emissions projections, SCP-M falls in the range of the CMIP5 models, although the oceanic carbon uptake is exaggerated for the RCP8.5 scenario. This result suggests that a more detailed experiment, for example with non-linear representation of the piston velocity with respect to atmospheric $CO_2$, or prescribed feedbacks from ocean circulation and biology (e.g. Meehl et al., 2007; IPCC, 2013a, b; Moore et al., 2018), might provide a closer fit to the CMIP5 models.

## 4    LGM-Holocene model-data experiment

### 4.1    Background

The LGM-Holocene transition, and glacial/interglacial variations in the carbon cycle in general, remain outstanding problems in paleoceanography (e.g. Sigman and Boyle, 2000; Kohfeld and Ridgewell, 2009; Hain et al., 2010; Ferrari et al., 2014; Kohfeld and Chase, 2017). At issue is the precise cause of 80-100 ppm variations in atmospheric $CO_2$ across glacial and interglacial periods. These $CO_2$ oscillations are accompanied by striking changes in ocean and atmospheric carbon isotopes,
oceanic carbonate ion distributions and other paleo chemical indicators. Of particular interest is the transition from the LGM,

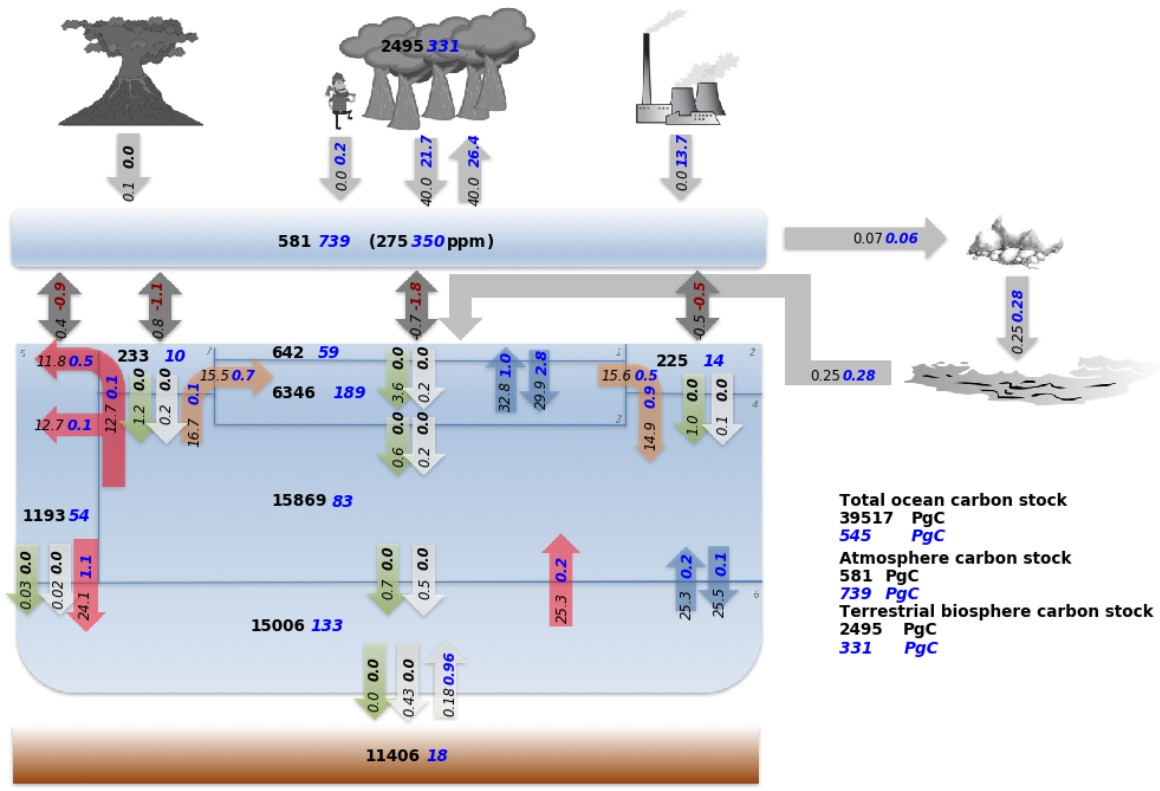

**Figure 9.** SCPM-modelled preindustrial carbon stocks and fluxes (in PgC in black text) compared with IPCC RCP6.0 emissions scenario by 2100 (shown as PgC changes with blue text for positive changes, red text for negative and black text = no change). Atmosphere, ocean and terrestrial biosphere take up the load of carbon from the industrial source. By 2100, carbon is fluxing into all ocean boxes, the terrestrial biosphere and continental sediment weathering/river fluxes. Preindustrial outgassing of $CO_2$ in the Southern Ocean is reversed, and carbon is returned to the ocean via enhanced $CaCO_3$ dissolution. Box numbers on the diagram refer to ocean regions specified in Fig. 1. Negative fluxes on bidirectional air-sea exchange arrows are fluxes of $CO_2$ out of the atmosphere into the ocean.

~18-24 ka (Yokoyama et al., 2000; Clark et al., 2009; Hesse et al., 2011; Hughes et al., 2013; Hughes and Gibbard, 2015), to the Holocene (11.7 ka-present), due to the growing abundance of proxy data covering that period. The causes of abrupt atmospheric $CO_2$ rise at the termination of the LGM, and continuing up to the Holocene period, remain definitively unresolved. The ocean is likely the main driver of atmospheric $CO_2$ on the relevant timescale, due to its relative size as a carbon reservoir (e.g.

Broecker, 1982; Sarmiento and Toggweiler, 1984; Kohfeld and Ridgewell, 2009; Sigman et al., 2010), alongside changes in the terrestrial biosphere stock of carbon (Francois et al., 1999; Ciais et al., 2012; Peterson et al., 2014; Hoogakker et al., 2016). Active theories within the ocean realm include changes in ocean biology (Martin, 1990; Watson et al., 2000; Martinez-Garcia et al., 2014), ocean circulation and mixing (Sarmiento and Toggweiler, 1984; Toggweiler and Sarmiento, 1985; Toggweiler, 1999; Curry and Oppo, 2005; Anderson et al., 2009; Kohfeld and Ridgewell, 2009; De Boer and Hogg, 2014; Menviel et al.,

2016; Muglia et al., 2018), sea ice cover (Stephens and Keeling, 2000), whole ocean chemistry (Broecker, 1982; Sigman et al., 2010), or composite hypotheses (Kohfeld and Ridgewell, 2009; Ferrari et al., 2014; Kohfeld and Chase, 2017). Other mechanisms proposed include ocean temperature, the terrestrial biosphere, ocean volume and shelf carbonates (Opdyke and Walker, 1992; Trent-Staid and Prell, 2002; Ridgewell et al., 2003; Ciais et al., 2012; Annan and Hargreaves, 2013). Each hypothesis listed above is supported by either of site-specific tracer observations (e.g. marine carbonate cores), regional data

aggregation and review, literature synthesis, or modelling. Within the spectrum of hypotheses, a simple explanation of a carbon cycle mechanism, or mechanisms, remains elusive. Many of the early hypotheses were presented as independent, or even competing in causality for the interglacial $CO_2$ variation (Ferrari et al., 2014).

Substantial progress has been made over the last fifteen years, in constraining the list of likely candidates to ocean physical and biological processes, likely in concert. The growth of paleo datasets (e.g. Oliver et al., 2010; Peterson et al., 2014; Yu

et al., 2014b; Skinner et al., 2017; Zhao et al., 2017), and improvements in computing power, have enabled model and model-data studies which seek to constrain the magnitude of changes in the carbon cycle across the glacial/interglacial cycles (e.g. Stephens and Keeling, 2000; Toggweiler et al., 2006; Tagliabue et al., 2009; Hain et al., 2010; Bouttes et al., 2011; Hesse et al., 2011; Tschumi et al., 2011; Menviel et al., 2016; Kurahashi-Nakamura et al., 2017; Muglia et al., 2018). For example, Menviel et al. (2016) modelled slowing GOC and AMOC, with a modest increase in biological productivity in the Southern Ocean

in the LGM, using $\delta^{13}C$ data and an intermediate complexity earth system model. This differed from the finding of Muglia et al. (2018), who specifically examined the AMOC and Southern Ocean biological productivity. They found a weaker AMOC and stronger biological productivity could account for the LGM and Holocene $\delta^{13}C$, $\Delta^{14}C$ and $^{15}N$ data. The GOC was not tested by Muglia et al. (2018). Kurahashi-Nakamura et al. (2017) contradicted both studies, diagnosing a more vigorous (but shallower) AMOC in the LGM, using a GCM with data assimilation of various proxies, notably only incorporating Atlantic

data for the LGM.

## 4.2    Model-data experiments

We illustrate SCP-M's capabilities by solving for the parameter values of best-fit with late Holocene and LGM ocean and atmosphere proxy data, using a comprehensive model results-data optimisation. For this illustrative example, the atmosphere

**Table 3.** Changes to ocean and atmosphere parameter settings in SCP-M to recreate the LGM background model state

| Indicator | LGM change |
|---|---|
| Surface ocean box temperatures | -5-6$^\circ$C (Trent-Staid and Prell, 2002; Annan and Hargreaves, 2013) |
| Surface ocean box salinity | +1.0 psu (Adkins et al., 2002) |
| Polar ocean box piston velocity | x0.3 (Stephens and Keeling, 2000; Ferrari et al., 2014) |
| Ocean surface area and volume | -3.0% (Adkins et al., 2002; Grant et al., 2014) |
| Atmosphere radiocarbon production | x1.25 (Mariotti et al., 2013) |

As shown in the sensitivity tests in Fig. 4, some processes do not exert a strong influence on atmospheric $CO_2$, but do impact modestly on $CO_2$ and strongly on $\delta^{13}C$ and $\Delta^{14}C$. Where these features are posited to vary around glacial cycles, we have incorporated them as a step change from late Holocene/modern estimates, in our LGM model experiment.

and ocean data is taken from published sources (Table 2), averaged for the LGM ($\sim$18-24 ka) and late Holocene (6.0-0.2 ka) time periods and for box coordinates in SCP-M for the ocean data (depth and latitude). The mean and variance for each box is then calculated in SCP-M. First, we probe the potential for key model parameters to drive Holocene-LGM changes in atmospheric carbon variables, to focus our experiment on these parameters. It is likely that the LGM to Holocene carbon cycle

changes were dominated by the ocean (Sigman and Boyle, 2000; Kohfeld and Ridgewell, 2009), but were also accompanied by a range of physical changes in the atmosphere and terrestrial biosphere that in aggregate, could be material (e.g. Sigman and Boyle, 2000; Adkins et al., 2002; Kohfeld and Ridgewell, 2009; Ferrari et al., 2014). These changes include sea surface temperature, salinity, sea-ice cover, ocean volume and atmospheric $^{14}C$ production rate. Estimates of average sea surface temperature for the LGM generally fall in the range of 3-8$^\circ$C cooler than the present (Trent-Staid and Prell, 2002; Annan and

Hargreaves, 2013). Adkins et al. (2002) estimated ocean salinity was 1-2 psu higher in the LGM and sea levels were $\sim$120m lower (Adkins et al., 2002; Grant et al., 2014). Stephens and Keeling (2000) and Ferrari et al. (2014) highlighted the role of expanded sea ice cover in the Southern Ocean during the LGM as a key part of the LGM $CO_2$ drawdown. Finally, Mariotti et al. (2013) estimated that higher atmospheric radiocarbon production accounted for $+\sim$200‰ in atmospheric $\Delta^{14}C$ in the LGM. Mariotti et al. (2013) simulated this variation in model experiments by increasing the radiocarbon production rate by a

multiple of 1.15-1.30 (best guess 1.25) of the modern estimate in order to recreate LGM $\Delta^{14}C$ values. Using these findings we define two background states for modelling purposes: a late Holocene state (Table 6 in the Appendix) and the LGM state (Table 3). Figure 10 shows the cumulative effect of these changes in SCP-M, within the late Holocene-LGM atmosphere 3D $CO_2$-$\delta^{13}C$-$\Delta^{14}C$ data space. These changes are the first stage of a model adjustment to analyse the potential for ocean circulation and biological changes to deliver the LGM atmospheric $CO_2$, $\delta^{13}C$ and $\Delta^{14}C$ values, and transition the model output from the

red circle (late Holocene) to the black star (the LGM background settings), and then to the black circle (LGM). The decrease in ocean surface box temperatures leads to a drop in $CO_2$ of $\sim$20 ppm and a lightening of $\delta^{13}C$ by $\sim$0.6‰, owing to the increased solubility of $CO_2$ in colder water, and the increasing fractionation of $\delta^{13}C$ with decreasing temperatures, which leaves more

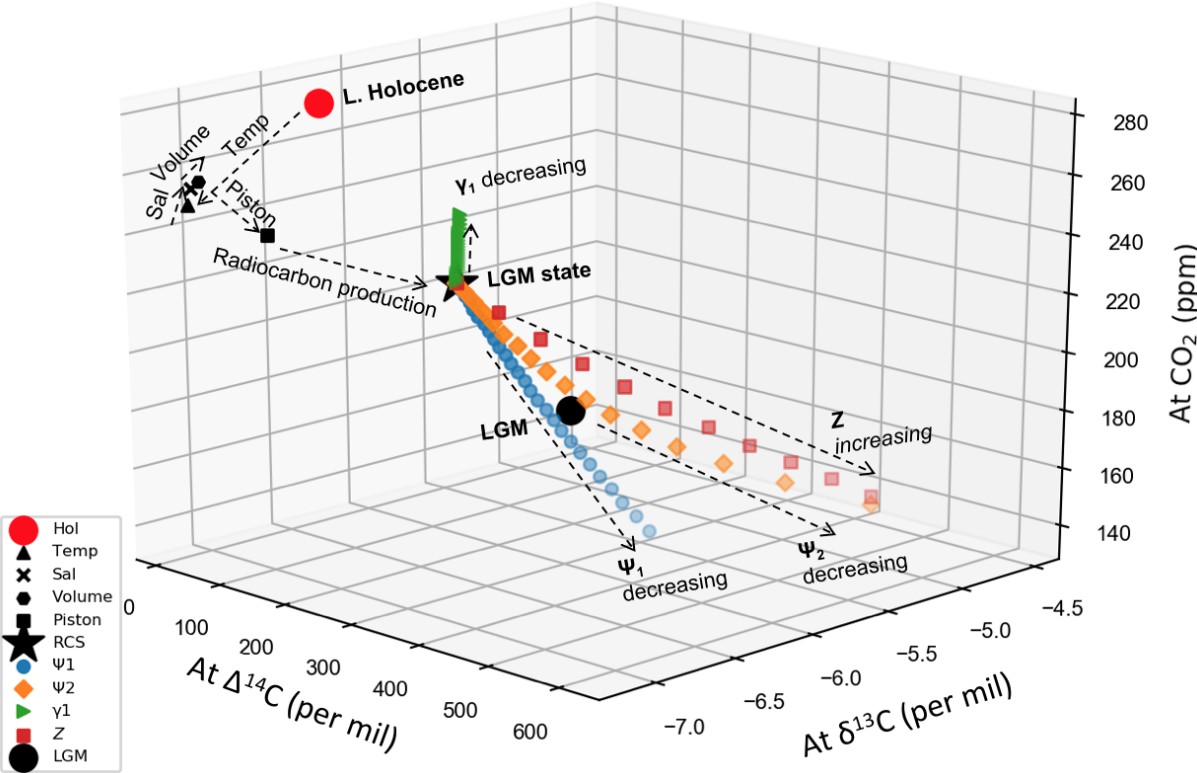

**Figure 10.** LGM state parameter adjustments. Using the posited LGM changes in environmental parameters in Table 3, we establish the LGM foundations for exploring the impacts of varying large scale ocean process parameters towards LGM atmospheric $CO_2$-$\delta^{13}C$-$\Delta^{14}C$ data space. The red circle is our starting point for the late Holocene. From the LGM state foundation (black star), variation of global overturning circulation ($\Psi_1$), Atlantic meridional overturning circulation ($\Psi_2$) and the soft-tissue biological pump ($Z$), drives atmospheric $CO_2$, $\delta^{13}C$ and $\Delta^{14}C$ into the vicinity of their LGM data values (black circle). The biological pump $Z$ can effect the LGM $CO_2$ outcome, but steers $\delta^{13}C$ away from the LGM value. Both $\Psi_1$ (3-29 Sv) and $\Psi_2$ (3-19 Sv) experiments run very close to the LGM data values on their own, although neither can deliver a precise hit.

$^{12}C$ in the atmosphere. There is limited impact on $\Delta^{14}C$. Increasing salinity slightly reverses these changes to $CO_2$ and $\delta^{13}C$. Reducing sea surface area and volume slightly increases $CO_2$ and increases $\Delta^{14}C$ as the ocean's capacity to take up these elements is reduced. Slowing down the piston velocity in the polar Southern Ocean box, as a proxy for increased sea ice cover, slightly reduces $CO_2$ (reduced outgassing), increases $\Delta^{14}C$ (slower rate of invasion to the ocean) and increases $\delta^{13}C$ (reduced outgassing and sea-to-air fractionation of $\delta^{13}C$). Finally, increasing the rate of atmospheric radiocarbon production forces a shift in $\Delta^{14}C$ (horizontal shift in Fig. 10) towards the LGM levels (black star and circle in Fig. 10). In aggregate, these changes lead to a fall in $CO_2$ of ~35 ppm, a fall in $\delta^{13}C$ of ~-0.5‰ and an increase in $\Delta^{14}C$ of ~300‰.

**Table 4.** Parameter value ranges for the late Holocene and LGM model-data experiments.

| Parameter (unit) | L. Holocene exp range | LGM exp range |
|---|---|---|
| $\Psi_1$ (Sv) | 20-35 | 15-30 |
| $\Psi_2$ (Sv) | 15-25 | 5-20 |
| $\gamma_1$ (Sv) | 15-30 | 5-35 |
| $Z$ (mol C m$^{-2}$ yr$^{-1}$) | 2-7 | 2-7 |

From the black star in Fig. 10, the "LGM state", we perform a focussed sensitivity test on key hypothesised drivers of LGM-Holocene carbon cycle changes (Kohfeld and Ridgewell, 2009; Sigman et al., 2010). These are: slower GOC ($\Psi_1$), slower AMOC ($\Psi_2$), reduced deep-abyssal ocean mixing ($\gamma_1$) and a stronger biological pump ($Z$). The $Z$ global biological production parameter, varied across 5-10 mol C m$^{-2}$ yr$^{-1}$ (i.e. increased), can deliver the LGM CO$_2$ changes, but steers $\delta^{13}$C and $\Delta^{14}$C away from their LGM values. $\gamma_1$ drives ancillary changes in all three variables, suggesting it is not the driver of the LGM atmospheric changes but may play a modulating role. Both $\Psi_1$ (3-29 Sv) and $\Psi_2$ (3-19 Sv) experiments run very close to the LGM data values on their own, although neither can deliver a precise hit.

Using the literature-referenced Holocene and LGM background parameter states, and informed by the sensitivity analysis in Fig. 10, we take advantage of SCP-M's fast run time to perform thousands of multi-variant simulations over the free-floating $\Psi_1$, $\Psi_2$, $\gamma_1$ and $Z$ parameter spaces, using the SCP-M batch module. We then perform an optimisation routine against the data for each period to solve for values for $\Psi_1$, $\Psi_2$, $\gamma_1$ and $Z$. The SCP-M batch module cycles through each set of parameter combinations, with each model simulation run for 10,000 years. Table 4 shows the experiment parameter ranges for the late Holocene and LGM model-data experiments.

The parameter input ranges for the experiments were informed by the sensitivity tests shown in Fig. 4 and Fig. 10. For example, the responses of atmospheric CO$_2$, $\delta^{13}$C and $\Delta^{14}$C to variations in $\Psi_1$, $\Psi_2$ and $Z$, lead us to cater for lower values for $\Psi_1$ and $\Psi_2$ (weaker ocean circulation) and higher values for $Z$ (increased biological productivity) in the LGM experiment. Where the experiments resulted in a parameter output value at the limit of the input range, the range was widened and the experiment was repeated. 16,896 and 47,616 simulations were undertaken for the late Holocene and LGM experiments, respectively.

The SCP-M script harvests model output and performs a least squares optimisation of the output against the LGM and late Holocene data for atmospheric CO$_2$, atmospheric and ocean $\Delta^{14}$C and $\delta^{13}$C, and also oceanic carbonate ion proxy, to source the best-fit parameter values for $\Psi_1$, $\Psi_2$, $\gamma_1$ and $Z$ (or any parameter specified):

$$Opt_{n=1} = Min \sum_{i,k=1}^{N} (\frac{R_{i,k} - D_{i,k}}{\sigma_{i,k}})^2 \tag{27}$$

**Table 5.** Late Holocene and LGM model-data parameter optimisation and associated atmospheric variable model output

| Parameter (units) | Data values L. Holocene (LGM) | late Holocene experiment results | LGM experiment results |
|---|---|---|---|
| $\Psi_1$ (Sv) | 20-30 (na) | **30** | **18** |
| $\Psi_2$ (Sv) | 15-25 (na) | **18** | **15** |
| $\gamma_1$ (Sv) | na (na) | **28** | **31** |
| $Z$ (mol C m$^{-2}$ yr$^{-1}$) | 2-10 (na) | **5** | **5** |
| At $CO_2$ (ppm) | 275±6 (195±3) | 275 | 194 |
| At $\delta^{13}$C (‰) | -6.35±0.09 (-6.46±0.01) | -6.35 | -6.46 |
| At $\Delta^{14}$C (‰) | 20±48 (414±32) | 21 | 404 |

Bold font parameter results indicate those parameters that are free-floating and determined by the model and data in the experiment. The LGM experiment shows a marked decline in the strength of global overturning circulation $\Psi_1$ (-12 Sv), and a modest decline in Atlantic meridional overturning circulation $\Psi_2$ to deliver the LGM atmosphere and ocean data signal. A minor increase in deep-abyssal mixing $\gamma_1$ is also seen.

where: $Opt_{n=1}$ = optimal value of parameters $n$, $R_{i,k}$ = model output for concentration of each element $i$ in box $k$, $D_{i,k}$ = average data concentration each element $i$ in box $k$ and $\sigma_{i,k}$ = standard deviation of the data for each element $i$ in box $k$. The standard deviation performs two roles. It reduces the weighting of data with high uncertainty and also normalises for the different unit scales (e.g. ppm, ‰ and umol kg$^{-1}$), which allows multiple proxies in different units to be incorporated in the optimisation. Where data is unavailable for a box, that element and box combination is automatically omitted from the optimisation routine.

The late Holocene data-optimised results for $\Psi_1$ (30 Sv) and $\Psi_2$ (18 Sv) show good agreement with the Talley (2013) observations for $\Psi_1$ (29 Sv) and $\Psi_2$ (19 Sv) from the the modern ocean (Table 5). The starting global value of $Z$, of 5 mol C m$^{-2}$ yr$^{-1}$, is returned in the experiment. The experiment also successfully returns values for atmospheric $CO_2$, $\delta^{13}$C and $\Delta^{14}$C within standard error for the late Holocene data series (Table 5).

The ocean and atmosphere SCP-M results for the LGM (bold stars) and late Holocene (transparent stars) experiments using the optimised parameter settings in Table 5, are plotted in Fig. 11 along with the corresponding data (blue dots with error bars). The experiment provides results within the error bounds of data for most of the box regions in both scenarios, and an excellent fit to the change in the relative distribution of the proxies between ocean boxes and the atmosphere which is preserved in the LGM and late Holocene data. A key feature of the ocean $\delta^{13}$C data is a depletion of deep ocean $\delta^{13}$C in the LGM, shown as a drop in $\delta^{13}$C values in the deep (box 4) and abyssal (box 6) boxes, relative to the intermediate box (3). In the LGM $\delta^{13}$C data, there is a spread of 1‰ across these water masses, which narrows to 0.3‰ in the late Holocene data. The pattern is replicated in the LGM model experiment, pointing to the important role of changes in abyssal-deep ocean water flows, via $\Psi_1$, in delivering the ocean $\delta^{13}$C data patterns. The model shift in $\delta^{13}$C in the deep box (box 4) of 0.6‰ from the LGM to late Holocene, is in

good agreement with a global deepwater estimate of 0.49 ±0.23‰ by Gebbie et al. (2015) and an earlier estimate of 0.46‰ by Curry et al. (1988). The average atmospheric $\delta^{13}C$ value remains largely unchanged between the two periods, due to the effect of the terrestrial biosphere, which causes net uptake $CO_2$ in the Holocene period (increases atmospheric $\delta^{13}C$), and net respiration of $CO_2$ in the LGM period (decreases atmospheric $\delta^{13}C$).

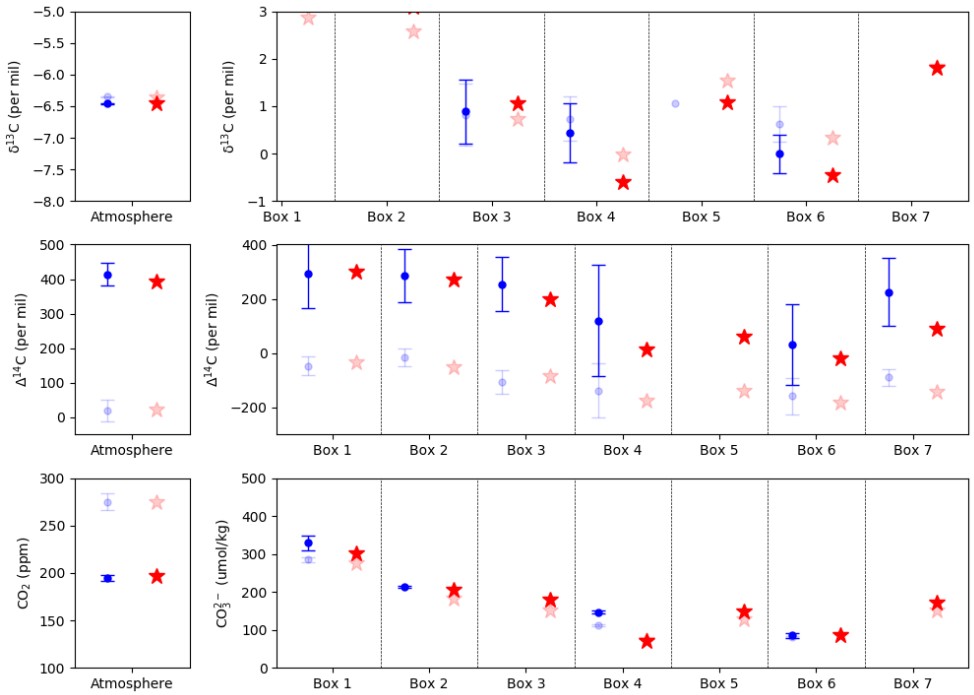

**Figure 11.** LGM atmosphere and ocean data-optimised model results. Left panels shows the atmospheric carbon cycle results from SCP-M (red stars) plotted against LGM average data values (blue dots) with standard error bars. The right panel shows the SCP-M ocean results plotted against LGM average ocean data where available. Corresponding Holocene data and results shown with transparent markers. The data-optimised model results show a close match for the LGM atmospheric data and most of the ocean data. The ocean $\delta^{13}C$ and $\Delta^{14}C$ data show an increased compositional gradient between shallow-intermediate depths (boxes 1-3) and deep-abyssal depths (boxes 4 and 6), an outcome replicated in the corresponding model results mainly by a slower GOC. Data sources are shown in Table 2.

The model results also closely replicate the reduction in deep-to-shallow ocean compositional gradient in $\Delta^{14}C$ data moving from the LGM to Holocene period (e.g. Skinner and Shackleton, 2004; Skinner et al., 2010; Burke and Robinson, 2012; Skinner et al., 2015; Chen et al., 2015; Hines et al., 2015; Ronge et al., 2016). The LGM data shows a spread of ∼300‰ between abyssal (box 6) and intermediate (box 3) waters, and deep (box 4) versus surface (boxes 1, 2 and 7) boxes. In the late Holocene data, the spread is narrowed to ∼100‰. This data observation was popularly characterised as the result of increased Southern Ocean

upwelling of $\Delta^{14}C$-depleted deep water into intermediate and shallow depths in the Holocene (e.g. Skinner et al., 2010; Burke and Robinson, 2012; Skinner et al., 2015). A slow-down in Southern Ocean upwelling in the LGM allows $\Delta^{14}C$-depleted

water to accumulate in the deep or abyssal ocean and a widening in the $\Delta^{14}$C gradient between deep and shallow waters. In SCP-M, this is simulated by lower values for $\Psi_1$ and $\Psi_2$. The low latitude surface box (box 1) enrichment in $\Delta^{14}$C in planktonic foraminifera in the LGM, is replicated by the increased atmospheric production rate of radiocarbon applied to the LGM experiment, combined with slower ocean circulation.

SCP-M results are shown for comparison with sparse carbonate proxy data (Fig. 11 bottom panel). The model results for the carbonate ion proxy mirror the limited variation in the data between the LGM and late Holocene. The changes are most pronounced in the surface boxes (boxes 1 and 2), which are under the influence of atmospheric $CO_2$, and attenuate somewhat in the deeper boxes (boxes 4 and 6). Yu et al. (2014b) interpreted the relatively small changes in carbonate ion in the deepest ocean (box 6) as the result of efficient buffering of deep water pH by carbonate dissolution, most notably in the Pacific Ocean. The model result for the deep box (box 4) goes against the LGM-Holocene variation in the data, but given there is only one data point for this part of the ocean, and the variation itself is small, it is an uncertain outcome.

The LGM scenario shows important changes in the carbon redistributive behaviour of the ocean (Fig. 12). The stock of carbon increases in abyssal and deep boxes (blue text denotes the increase in PgC from late Holocene to LGM), and reduces in the intermediate, low latitude surface and northern surface boxes (red text denotes the decrease in PgC from late Holocene to LGM). The amount of carbon upwelled to the subpolar surface and deep boxes by GOC ($\Psi_1$), drops by $\sim$ 5-10 PgC yr$^{-1}$, with the most pronounced changes taking place at the abyssal-deep box boundary. The slower upwelling rate of carbon causes a reduced outgassing rate of $CO_2$ from the subpolar box to the atmosphere. The weaker flux of $\Psi_2$ also brings a reduced DIC load into the intermediate depth ocean, the driver for lower DIC content in the intermediate and surface boxes. The optimised parameter run for the late Holocene results in a terrestrial biosphere carbon pool of 2,495 Pg C, which is fortuitously close to the preindustrial estimate of Raupach et al. (2011) (2,496 Pg C), at the top end of acceptable values in Francois et al. (1999), and close to the "active" land carbon pool of 2,370 $\pm$ 125 estimated by Ciais et al. (2012). In the optimised LGM model results, the terrestrial biosphere is reduced by 667 Pg C from the late Holocene value, which is towards the upper bound of recent estimates of this change (0 - 700 Pg C e.g. Ciais et al. (2012), Peterson et al. (2014)), but within uncertainty bounds. For example, Peterson et al. (2014) estimated a variation of 511 $\pm$ 289 Pg C in the terrestrial biosphere carbon stock based on whole of ocean $\delta^{13}$C data, the same data used in this exercise. According to Francois et al. (1999), palynological and sedimentological data infer that the terrestrial biosphere carbon stock was 700-1350 PgC smaller in the LGM than the present. Ciais et al. (2012) pointed to a growth of a large inert carbon pool in steppes and tundra during the LGM, which may have modulated some of the active biosphere carbon signal (i.e. reduced terrestrial biosphere), a factor not explicitly covered in our modelling exercise. The terrestrial biosphere is clearly a key part of the LGM-late Holocene carbon cycle transition. The atmosphere-enriching fractionation of $\delta^{13}$C by the terrestrial biosphere during the deglacial period, effectively reverses the effects of the release of $\delta^{13}$C-depleted carbon from the deep ocean to the atmosphere at the termination and leaves atmosphere $\delta^{13}$C almost unchanged from LGM values as a result (Schmitt et al., 2012). The DIC:Alk balances in the abyssal ocean during the LGM also drive subtle changes in the balance of carbonate out-flux by sinking and influx from sediment dissolution, which build up to substantial differences in the sediment carbon stock between the LGM and Holocene simulations, mainly due to the timeframes modelled in the SCPM spin-up for each scenario (15 kyr).

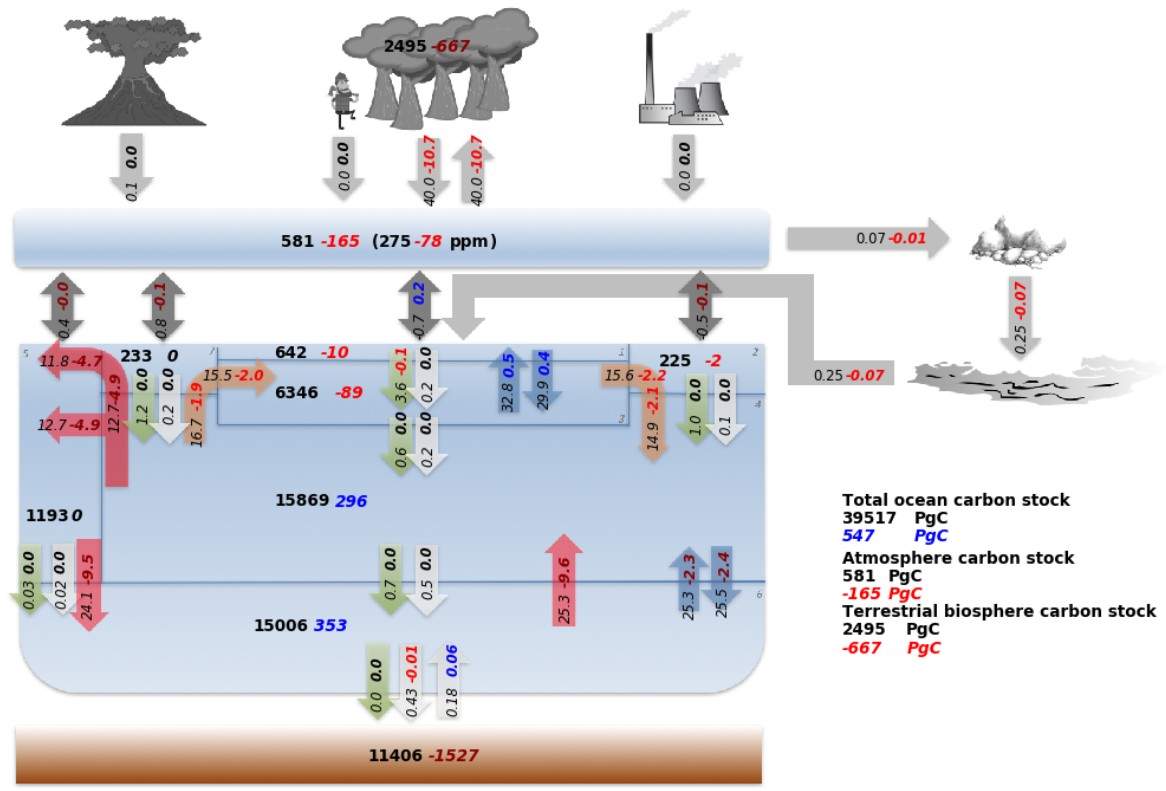

**Figure 12.** Late Holocene (figures in black text) and LGM (shown as PgC changes from the late Holocene) carbon stocks and fluxes modelled with SCP-M. For the LGM blue text shows positive changes (in PgC), red text shows negative and black = no change. LGM parameter values selected from the 4-parameter LGM experiment in Table 5. The LGM setting leads to a transfer of carbon from the atmosphere and terrestrial biosphere to the deep ocean. Carbon upwelled into the surface ocean falls, leading to reduced outgassing of $CO_2$ in the Southern Ocean boxes. Continental weathering and river fluxes of carbon are also reduced due to lower atmospheric $CO_2$, leading to a change in $CaCO_3$ burial and dissolution in marine sediments until equilibrium is restored with river input to the oceans. Box numbers on the diagram refer to ocean regions specified in Fig. 1.

## 5 Discussion

### 5.1 Model advantages and limitations

In this paper we introduce SCP-M, a box model of the global carbon cycle. We demonstrate its application to the modern and future carbon cycle with anthropogenic emissions, and reconstructing potential changes across the LGM-late Holocene carbon cycle transition. In summary, SCP-M is a simple, easy to use model of the carbon cycle, and its fast run time enables comprehensive scenario analysis or optimisations for scenario or hypothesis-testing. It takes approximately 30 seconds to complete a 10,000 year simulation, making the model useful for long-term paleo- reconstructions of the carbon cycle. Our

LGM-late Holocene experiment (Section 4) includes broad variations in GOC, AMOC, deep-abyssal mixing and global biological productivity. The experiments cover up to ~47,000 parameter combinations across the LGM and late Holocene proxy data, reducing the possibility of confirmation bias or predetermined outcomes. Furthermore, the model's simplified topology (Fig. 1), albeit consistent with an observationally-based understanding of the ocean (e.g. Talley, 2013), makes it accessible to a wide user-group and potentially useful as a teaching aid to illustrate high-level concepts in the carbon cycle. The model contains

data modules that directly integrate data via box-mapping and averaging processes for use in calibration and for model-data experiments (Fig. 2). It also includes a model-data optimisation routine to elicit parameter values that best fit the data, described in Section 4.2.

A limitation of SCP-M v1.0 is that it does not distinguish between the Atlantic/Indo-Pacific ocean basins, which is a large simplification. We argue that this is feasible for testing high-level hypotheses, for example involving large-scale ocean processes

across the LGM/Holocene time periods, and the model is demonstrated to produce appropriate results in such an application. However, this framework may not be useful for testing localised or detailed problems. Given SCP-M is a box model, there are other simplifications, including a rigid and perhaps somewhat arbitrary treatment of box boundaries (Fig. 1). For some experiments the box boundaries themselves may need to be a dynamic, model-determined output. In our LGM-Holocene example, we didn't vary the abyssal box thickness across the time periods. However, this could be done very easily for the

case of an experiment featuring shoaling or deepening water mass boundaries (e.g. Curry and Oppo, 2005). A key drawback of the model is that it can identify the cause of changes in proxy element concentrations, in terms of parametrised processes, but cannot diagnose the root cause of these process changes. For example, with SCP-M we cannot directly answer the question of what causes GOC, AMOC or biological productivity to vary on glacial/interglacial cycles, but combined with data we can propose which of these varies (Section 4.1).

### 5.2 Modern carbon cycle simulations

Our simple forcing of SCP-M with historical and projected anthropogenic $CO_2$ emissions and SST (Section 3.3, Figs. 6-8), shows that SCP-M can reproduce historical data and the model results of more complex CMIP5 models. The SCP-M results for atmospheric $CO_2$ (Fig. 6), air-sea fluxes of carbon (Fig. 7) and accumulation of carbon in the various carbon reservoirs

5  (Fig. 8), line up in the range of CMIP5 model projections. Importantly, SCP-M is shown to replicate the historical data over the period 1751-2016 for atmospheric $CO_2$, $\delta^{13}C$ and $\Delta^{14}C$ (Figs. 5 and 6). The historical period is a stringent test of carbon cycle

models because it incorporates the influences of anthropogenic emissions, atmospheric bomb testing, and an abundance of data observations of the Earth's carbon cycle response for comparison. The radioactive decay and dispersal of bomb-produced $^{14}$C, provides an excellent 'time clock' for the fluxes in the carbon cycle, particularly air-sea gas exchange and ocean circulation. Our experiment incorporates forcing of atmospheric $^{14}$C during 1954-63, and SCP-M appropriately replicates the take-up of bomb $^{14}$C by the ocean from the atmosphere in the following years (Fig. 5).

However, the SCP-M modern/future simulations are simple, and fail to take account of more complex, potential feedbacks in the carbon cycle. These may include a wind shift-induced slowing of AMOC and thermocline mixing, or a response of ocean biological productivity to changed pCO$_2$, temperature and DIC in the surface ocean (e.g. Meehl et al., 2007; IPCC, 2013a, b; Moore et al., 2018). To simulate such feedbacks, the relevant parameters would need to be forced in SCP-M. A dynamical response would be expected in more complex Earth system models. The value of SCP-M is in rapidly undertaking 'what-if' type analysis, to probe the effects of such changes under a variety of scenarios. For example, the high-level testing of negative emissions processes such as alkalinity or iron seeding of the ocean, rock waste fertilisation and afforestation/reforestation on land, or marine fauna management, as tools for reducing atmospheric CO$_2$ on a global scale, are feasible uses of SCP-M.

## 5.3   LGM-late Holocene modelling

Our model-data optimisation using SCP-M and published data (Section 4.2) showed that variations in the strength of the large scale ocean physical processes, particularly GOC and AMOC, can account for the LGM to Holocene carbon cycle changes inferred in the proxy data (Table 5). Critically, the variations are accompanied by changes in SST, sea-ice cover and the terrestrial biosphere (Table 3). The result is observed on account of ocean and atmosphere data across the proxies of CO$_2$, $\delta^{13}$C, $\Delta^{14}$C and carbonate ion (Fig. 11). This finding is not a new one, but corroborates the model-data conclusion of Menviel et al. (2016), box modelling of Toggweiler (1999) and $^{14}$C proxy data findings of Sikes et al. (2000) and Skinner et al. (2017). The importance of GOC is at odds with Muglia et al. (2018), who found for a substantially weakened AMOC and enhanced biological productivity in the Southern Ocean in the LGM, with no role examined for GOC in that study. Kurahashi-Nakamura et al. (2017) had an altogether different finding, modelling a stronger yet shallower AMOC during the LGM. Many such studies focus exclusively on the Atlantic Ocean, perhaps due to the well-understood AMOC and the more detailed proxy data coverage in that basin. For example, Curry and Oppo (2005) provided striking transects of $\delta^{13}$C in the LGM and late Holocene Atlantic Ocean, which provided evidence for large changes in the basin $\delta^{13}$C stratigraphy across the two time periods. The $\delta^{13}$C data compilations of Oliver et al. (2010) and Peterson et al. (2014) are also heavily skewed towards the Atlantic basin.

Talley (2013) emphasised the importance of the Pacific and Indian Oceans' overturning circulation limbs in the global ocean circulation regime, which implies that it is a critical part of the Earth's carbon cycle. This finding was corroborated by Skinner et al. (2017) in a recent review of Pacific Ocean radiocarbon data. The model-data results using SCP-M suggest that GOC was substantially reduced during the LGM (Table 5), accompanying enhanced storage of isotopically-depleted carbon in the abyssal and deep ocean from atmospheric and terrestrial biosphere sources (Fig. 11). We posit that the release of volumes of carbon, that were greater than amounts stored in the deep/abyssal Atlantic alone, caused the atmospheric CO$_2$ increase at the last glacial termination. Such a movement of carbon to/from the global abyssal ocean, is likely required due to the

large, opposite movement in atmospheric $CO_2$ from the terrestrial biosphere across the same time period (Fig. 12). During the LGM, the terrestrial biosphere was reduced relative to the modern period, which was a source of $CO_2$ to the atmosphere, and rebounded from the LGM to the Holocene, becoming a sink of $CO_2$ during that period (Francois et al., 1999; Ciais et al., 2012; Peterson et al., 2014; Hoogakker et al., 2016). Incorporating the terrestrial biosphere in the modelling experiments, increases the magnitude of carbon uptake/release required from the ocean to satisfy the LGM and late Holocene atmospheric $CO_2$ and critically $\delta^{13}C$ data (even when incorporating SST, salinity, sea-ice cover proxy and ocean volume changes). The finding underscores the importance of incorporating multiple data-proxies and carbon reservoirs in glacial/interglacial carbon cycle modelling.

Our model-data experiments did not find for a role of changed marine biological production in the LGM/late Holocene transition (Table 5). However, this finding was the result of testing for variations in the global value of the ocean biological productivity, impacting on all surface ocean boxes in SCP-M. Other studies (e.g. Menviel et al., 2016; Muglia et al., 2018) focussed specifically on the Southern Ocean biological productivity and identified its potential role in the LGM atmospheric $CO_2$ drawdown. The Southern Ocean marine biology, in particular is posited as a candidate for driving glacial/interglacial cycles of $CO_2$ (e.g. Martin, 1990; Martinez-Garcia et al., 2014).

## 6   Conclusions

The SCP-M carbon cycle box model was constructed for the purposes of scenario or hypothesis testing (quickly and easily), model-data integration and inversion, paleo reconstructions, and analysing the distribution of anthropogenic emissions in the carbon cycle. The model contains a full ocean-atmosphere-terrestrial carbon cycle with a realistic treatment of ocean processes. Despite being relatively simple in concept and construct, SCP-M can account for a range of paleo and modern carbon cycle observations. The model applications illustrated here include integration with datasets from the present day (GLODAPv2, IPCC) and ocean paleo proxy data across the LGM and late Holocene periods. Simulations of the modern carbon cycle indicate that SCP-M provides a realistic representation of the dynamic shocks from human industrial and land use change emissions and bomb $^{14}C$. A model-data experiment using LGM and late Holocene $CO_2$, $\delta^{13}C$, $\Delta^{14}C$ and carbonate ion proxy, is able to resolve parameter values for ocean circulation, mixing and biology while reproducing model results that are very close to the proxy data for both time periods. The results indicate that the LGM to Holocene carbon cycle transition can be explained by variations in the strength of global overturning circulation and Atlantic meridional overturning circulation, when combined with a number of background changes such as sea surface temperature, salinity, sea-ice cover, ocean volume and a varied atmospheric radiocarbon production rate. Further work on data quality and analysis is required to validate this finding, which is the subject of a separate paper. The results show promise in helping to further resolve the LGM to Holocene carbon cycle transition and point towards an ongoing application for data-constrained models such as SCP-M.

## 7 Code availability

The full model code and all file dependencies, with user instructions are located at: https://doi.org/10.5281/zenodo.1310161

## 8 Data availability

No original geochemical data was created in the course of the study, but any data compiled and used to run the model and model-data experiments is located with the model code at: https://doi.org/10.5281/zenodo.1310161

*Author contributions.* CO undertook model build, data-gathering, modelling and model-data experiments. AH provided the oceanographic interpretation, supervised model design, modelling work and designed the model-data experiments. ME provided input into model design, designed model-data experiments and oversaw the modelling of the marine biology and isotopes. BO provided input into model design, oversaw the modelling of carbonate chemistry, marine sediments and interpretation of LGM-Holocene hypotheses. SE designed model-data experiments and oversaw the modelling of the marine biology and carbonate pump. All authors contributed to drafting and reviewing the document.

*Competing interests.* The authors declare that they have no conflict of interest.

*Acknowledgements.* This manuscript was substantially improved by input from two anonymous reviewers and the GMD Topical Editor. Stewart Fallon provided guidance and spreadsheet tools for the processing of radiocarbon data. Malcolm Sambridge provided input on model-data optimisation and inversions. Jimin Yu provided helpful discussions and carbonate ion proxy data.

## Appendix A: Parameters, data sources and dimensions

**Table 6.** SCP-M model dimensions, model parameter starting values and starting data used for model spin-up.

| Model item | Value | Source |
|---|---|---|
| Ocean surface area ($m^2$) | $3.619 \times 10^{14}$ | https://www.ngdc.noaa.gov/mgg/global/etopo1_ocean_volumes.html |
| Average ocean depth (m) | $4,000$ | https://www.ngdc.noaa.gov/mgg/global/etopo1_ocean_volumes.html |
| Mass of the atmosphere (kg) | $5.1 \times 10^{18}$ | https://nssdc.gsfc.nasa.gov/planetary/factsheet/earthfact.html |
| Mean molecular weight of atmosphere (moles gram$^{-1}$) | 28.97 | https://nssdc.gsfc.nasa.gov/planetary/factsheet/earthfact.html |
| Temperature and salinity of the ocean | Various | GLODAPv2 dataset (https://www.nodc.noaa.gov/ocads/oceans/GLODAPv2/) |
| Modern ocean element concentrations | Various | GLODAPv2 dataset (https://www.nodc.noaa.gov/ocads/oceans/GLODAPv2/) |
| $\Psi_1$ global overturning circulation (Sv) | 29.0 | Talley (2013) |
| $\Psi_2$ NADW overturning (Sv) | 19.0 | Talley (2013) |
| $\gamma_1$ abyssal-deep mixing parameter (Sv) | 19.0 | Talley (2013) |
| $\gamma_2$ thermocline mixing (Sv) | 40 | Toggweiler (1999) |
| $Z$ biological soft carbon productivity @ 100m (mol C m$^{-2}$ yr$^{-1}$) | $1-6$ | Martin et al. (1987) |
| Martin b scalar value | 0.75 | Berelson (2001) |
| Air-sea exchange velocity (m day$^{-1}$) | 3.0 | Toggweiler (1999) |
| $^{13}$C air-sea fractionation factors | $0.9989 - 0.999$ | Mook et al. (1974) |
| $^{14}$C air-sea fractionation factors | $0.98 - 0.998$ | Toggweiler and Sarmiento (1985) |
| $^{13}$C "thermodynamic" air-sea factor | 0.99915 | Schmittner et al. (2013) |
| $^{14}$C "thermodynamic" air-sea factor | 0.999 | Toggweiler and Sarmiento (1985) |
| Organic $\delta^{13}$C fractionation factor | $\sim 0.975$ | Toggweiler and Sarmiento (1985) |
| C/P in org "Redfield ratio" | 130 | Takahashi et al. (1985) |
| Rain ratio (carbonate:org in sinking particles) | 0.07 | Sarmiento et al. (2002) |
| CaCO$_3$ dissolution rate (units day$^{-1}$) | 0.38 | Hales and Emerson (1997) |
| n order of CaCO$_3$ dissolution reaction rate | 1.0 | Hales and Emerson (1997) |
| K$_{sp}$ solubility coefficient for calcite | Various | Mucci (1983) |
| Carbon chemistry solubility and dissociation coefficients | Various | Weiss (1974), Lueker et al. (2000) |
| Atmosphere radiocarbon production rate (atoms s$^{-1}$) | $\sim 1.6$ | Key (2001) |
| Suess and bomb radiocarbon corrections | Various | Broecker et al. (1980), Key (2001), Sabine et al. (2004), Eide et al. (2017) |
| Radiocarbon decay rate (yr$^{-1}$) | 1/8267 | Stuiver and Polach (1977) |
| Volcanic emissions flux CO$_2$ (mol C yr$^{-1}$) | $5\text{-}6 \times 10^{12}$ | Modified from Toggweiler (2008) |
| River phosphorus flux (Tg yr$^{-1}$) | 15.0 | Compton et al. (2000) |

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
