# Peer review of "The [simple carbon project] model v1.0"

_Geoscientific Model Development, 2018_

## Referee Comment (RC1) · Anonymous Referee #1 · 23 Aug 2018

O'Neill et al., present a new simple carbon cycle model, mostly focused on the ocean. The model comprises an atmospheric box, 7 boxes in the ocean and 2 boxes for terrestrial carbon. The model also includes a simple weathering parametrization and can be forced by volcanic and anthropogenic emissions of carbon. The manuscript includes a description of the model, as well as a demonstration of the model performances under late Holocene and LGM conditions as well as for the RCP6.0 scenario. Please find below some comments that should be addressed before publication.

Main comment: There is a lot of different topics/issues presented in this paper (e.g. model description and concept, LGM pCO2 change, partition of carbon under anthropogenic forcing), however I would have liked to see additional information on the model experiments as well as more background information. The model description is incomplete without information on temperature, salinity and the carbon isotopes section should be moved to the main text. Sensitivity studies are performed but the initial set

of parameters are unclear and the reasoning behind the changes to these parameters is not substantiated, leaving the reader guessing as to why such experiment was performed and figuring out whether the range of parameters studied made physical sense or not.

1) Introduction The introduction focuses on glacial/interglacial variations in atmospheric CO2. This is indeed one part of the study, but not only. I would have thought that (at least) the first part of the introduction should be devoted to the reasoning behind setting up such a box model. P1, L.18: Despite years of research, and significant progress, the sequence of events leading to glacial/interglacial changes in atmospheric CO2 is still poorly constrained. However, I don't think this can be called the "LGM Holocene dilemma". And I think the authors mean "glacial/interglacial" variations and not "interglacial" (here and throughout the text, e.g. p2, L.4). P1, L.22: I am not sure these two references are the best to define the "LGM" P1, L. 26: and to the fact that the terrestrial carbon content was most likely reduced (e.g. Ciais et al., 2012, Peterson et al., 2014). P2, L.2: only the reference to one review (Sigman et al. 2010) is given, while additional references could be given for all the hypotheses cited (at least one per mechanism). Another review could be mentioned: Kohfeld and Ridgwell, 2009.

P2, L.4-11: I would strongly suggest to significantly revise this paragraph, which really does not do justice to the last 15 years of work on the topic of glacial/interglacial changes in atmospheric CO2. Many sensitivity experiments and transient simulations have been performed with box models, models of intermediate complexity and OGCMs to understand glacial/interglacial changes in pCO2. A few references (non-exhaustive list) include Stephens & Keeling (2000), Toggweiler et al., (2006), references within Kohfeld and Ridgwell (2009), Hain et al., (2010), Tagliabue et al., (2010), Hesse et al., (2011), Bouttes et al., (2012), Tschumi et al., (2011), Chikamoto et al., (2012), Menviel et al., (2012), Ganopolski & Brovkin (2017), Menviel et al., (2017). …. Many of which (if not all of them) also included a thorough model-data comparison.

On the contrary, I would have liked to see in the introduction more details with respect

to the rationale of constructing a new carbon cycle box model. P2, L. 25: Please reformulate "extra-ocean" (Please also reformulate header of section 2.4)

2) Model description The model description is incomplete. In section 3, it is stated that the model is forced by SST and SSS, however there is no mention of the treatment of temperature and salinity in the model. There is no description of the parametrization of the carbon isotopes in the main part of the manuscript. Since the manuscript focuses on carbon isotopes, the main formulations have to be clearly laid out. In addition, marine export production is prescribed (p9), but there is little information on the values used, how they were chosen and how they vary in the experiment. Figure 4 could be helpful in that sense: the late Holocene and/or modern day values of all parameters should be clearly indicated in that figure.

Specific comments:

P 3, L. 5-6: "simulates sources and sinks". Some of these sources and sinks are really simplified, for example anthropogenic and volcanic emissions are a simple prescribed flux into the atmosphere. Weathering and river fluxes are also close to a simple pre-scribed flux. So, for some it might be more precise to state "includes forcing" than "simulate sources".

P3, L. 13-15; I am confused by this sentence

Ocean circulation and mixing: Box 4: why is there no exchange with boxes 3, 5 and 7 in equation 1? From the matrix, it looks like there are exchanges with boxes 5 and 7 but not 3, why? Box 1: why no exchange with boxes 2 & 7 in equation 2?

P11, L. 15-17: "around glacial cycles" is not precise enough. In addition, I don't think this sentence is correct, as changes were opposite in the Atlantic and Pacific Oceans.

3) Modelling results P16, L. 17: Is [CO3] approximated by ALK-DIC or fully calculated using ALk, DIC, T, S, P? P16, L.20: please reformulate as "remineralization of organic matter" P17, L. 2-7: Please explain your reasoning behind varying the rain ratio . I don't

understand why changing the rain ratio impacts atmospheric D14C and I suppose that the surface ocean pCO2 change could eventually impact atm d13C, but not "heavily" (L. 6-7). P19, L. 6: Please add "and there is a reduced outgassing of old low D14C waters." P19, L. 8: please remove "around the interglacial cycles." And please note that the year of the ref is actually 2008 (Toggweiler, 2008).

P19, L. 9-14: I suppose the authors expect a change in pCO2 due to the change in ocean area resulting from varying sea-level (and thus ocean volume) on G/IG timescales. Please spell it out. Please take out "volume" on L.9. The impact on D14C is surprising though.

P22, L. 7: This sentence is not correct. Please reformulate. P22, L. 10-14: I don't really agree with this paragraph. It is probably true for simple carbon cycle box model for which all parameters have to be tested and therefore the G/IG CO2 problem is explored by assessing the impact of each parameters. But, over the last years the G/IG CO2 problem has also been studied with coupled models providing a representation (granted this representation is associated with large uncertainties) of physical and biological changes occurring during glacial times. P23, L4-6: I am not sure what is meant here or what has been done.

4) Discussion A discussion of the capacity of the model and the results is missing. I would have liked to see a paragraph on why this model should be used. What are its benefits and limitations? Its fast processing time should be discussed there too (instead of the introduction). I would have liked to see the results of the future experiments discussed in the context of the CMIP5 results. Only Jones et al. (2013) and Wang et al. (2016) are referenced for this part. I would have liked to see a discussion of the results of the LGM experiments in comparison with other studies. Recently Muglia et al., (2018 EPSL) studied the impact of ocean circulation and iron fertilization on the LGM oceanic carbon and on its carbon isotopic distribution. Menviel et al. (2016)'s conclusion was also consistent with your conclusion: a slightly weaker AMOC and a weaker global overturning circulation. Among others these two studies could help

discuss the effects of Z, psi1 and psi2 shown in Fig. 10.

Abstract: The second part of the abstract focuses on the LGM simulations. I would suggest to tone down that part and instead add some information about the use and limitations of the model.

Minor and typos: P3, L. 10-12: please reformulate P16, L. 29-35: please reformulate this paragraph. P17, L.10: "decreases" P19, L. 19: please reformulate this sentence P19, L. 26: Maybe "appropriate" instead of "accurate"
* * *

---

## Referee Comment (RC2) · Anonymous Referee #2 · 19 Dec 2018

This paper is mostly well-written. The model is nicely described and is a genuinely useful framework for investigating physical and biogeochemical controls on the marine carbon cycle. I have no serious concerns with the work and my comments are mostly suggestions for rewording and clarification. A few slightly more important issues are below, followed by line-by-line notes.

1. I did not see a description of the numerical method employed or confirmation of model stability and potential for numerical error. Fig 7 shows some potential error propagation / numerical oscillation? Have the authors investigated this?

2. Figure placement needs significant improvement. For example, Figure 9 appears 5 pages after it is mentioned and is in a different section.

3. A few times it is noted that the model supports a physical overturning mechanism for driving LGM-Holocene changes, it should be made clearer that this idea has been

proposed before and the current work supports it, rather than introducing the concept.

Line-by-line: Page2: line 15: summary of box models is too vague Line 32: "simple carbon project model model"

Page 3: Line 20: does 'zonally averaged' make sense? There are no spatial dimensions here? E.g. later zonally-averaged refers to a 2D model. Figure 1. It is not always clear which arrows exchange with which boxes. E.g. some arrows are entirely within one box, some cross box boarders but do not terminate. The diagram should show what is actually happening in the model.

Page 5 Line 14 to end: A little confusion over model dimensionality. Be precise here. Explain how the model has no spatial dimensions but does have a representation of sizes and locations of boxes (if that is indeed correct).

Figure 2. Caption – "implemented". Also explain the direction of arrows here.

Page 9: Line 1: Biological flux "parameters" or "parameterization" Line 2: "action of biological activity" – reword Line 21: sub-surface or subsurface

Page 11: Line17: 'lending it some interest', consider rewording Line 34: should be "non-saturation-dependent" ?, "Earth"

Page 13: line 5: replace "extra ocean" with something more descriptive line 22: non-zero line 25: short-term, long-term

Page 14: Line 3: state meaning of beta parameter Figure 4: This is hard to read, consider better ways to display (e.g. title and unit on x axis)

Page 19: Line 7: "the carbon isotopes" reword Line 12 "the values for the isotopes" too vague Line 17: "response to the shocks" does not give the right impression

Figure 5. Explain data in panel b, why are there two data lines for the atmosphere? Remove "selection of boxes shown to reduce clutter", ironically this statement is itself clutter.

Figure 6. Remove "fed into" from caption. Use inputted or similar.

Page 20: Line 17: "carbon cycle destination for human emissions" – not clear what this means Line 19-20: explain this in more detail, a little confusing Line 22: the figures are becoming a long way from the relevant text by this point in the paper.

Figure 7: use of multiple transparencies and colors here makes it very hard to see the ranges, especially in the bottom panel. Also, it appears there is some oscillation developing in the model? Have you investigated this?

Figure 8: This is quite simple, can a comparison be incorporated? Line 22: "release of emissions" should be reworded

Figure 9: I would consider if there is a better way to show this. It takes a very long time to decode this information. Bar or pie charts would be more easily understood.

Page 22: Line 8: first line here needs clarification.

Page 25: Note that table 4 is in the appendix

Page 29 Line 8: net respiration versus net uptake should be made clear

Page 30 Line 10: don't need 'however' here Line 30: "showed deltas in the range of" – be more precise here

Page 31 Line 6: "but critically *are* accompanied" Line 7: *the* carbonate ion proxy? Line 23: superscript 14C Line 25: it cannot be wholly explained by overturning changes, as these must be combined with temp/salinity etc. changes listed afterwards. Also this work confirms that reduced overturning can drive the changes, rather than proposing this.

Figure 12: as with Figure 9, it's very difficult to get any meaning from this figure. Perhaps colours could be used to denote increases/decreases at the very least?

---

## Author Comment (AC1) · 6 Feb 2019

Thank you for your constructive and thorough comments, suggestions and input into the manuscript. We feel it makes a very strong contribution to the quality of the work. Please see below our responses to the individual comments. We have made reference to changes to the manuscript, which is included as a supplement to the author comments, in track changes.

Page and line references below refer to locations in the revised document with track changes. Please note the attached, marked-up document contains amendments from both sets of reviewer comments.

RC General Comment: There is a lot of different topics/issues presented in this paper (e.g. model description and concept, LGM pCO2 change, partition of carbon under anthropogenic forcing), however I would have liked to see additional information on the

model experiments as well as more background information. The model description is incomplete without information on temperature, salinity and the carbon isotopes section should be moved to the main text. Sensitivity studies are performed but the initial set of parameters are unclear and the reasoning behind the changes to these parameters is not substantiated, leaving the reader guessing as to why such experiment was performed and figuring out whether the range of parameters studied made physical sense or not.

AC: We have addressed these comments in more detail in response to the specific comments below. As a general comment, we have not tried to exhaustively review or document the starting values for all parameters. However, in response to the comment we have added additional text in Section 2.2.2 (Ocean and circulation and mixing) to explain our choice of parameters for the modern/late Holocene model spin-up. In response to the comments, we have also added more detail to Section 2.2.3 (Biological flux parameterisation) to explain our input values for marine biological production/export parameters. Throughout the document we have added more references to Table 6 in the Appendix that shows the model's parameters and dimensions, and their sources. At the start of Section 3.2 (Sensitivity tests), we have added a paragraph to explain the rationale for undertaking the sensitivity tests, and what range of values we have chosen. In addition, as suggested in the comments below, we have added to the Figure 4 subplots the modern parameter values/assumptions for visual reference with the sensitivity tests.

1) Introduction

RC: The introduction focuses on glacial/interglacial variations in atmospheric CO2. This is indeed one part of the study, but not only. I would have thought that (at least) the first part of the introduction should be devoted to the reasoning behind setting up such a box model.

AC: We have re-arranged the introduction by moving the discussion of box models and

**[GMDD](https://www.geosci-model-dev.net/)**
rationale for SCP-M, to the front (Page 2, line 16). We have moved the discussion of the LGM-Holocene modelling to a later section in the paper. For this reason, many of the following items can now be found in section 4 (Page 30, line 3).

RC: P1, L.18: Despite years of research, and significant progress, the sequence of events leading to glacial/interglacial changes in atmospheric CO2 is still poorly constrained. However, I don't think this can be called the "LGM Holocene dilemma". And I think the authors mean "glacial/interglacial" variations and not "interglacial" (here and throughout the text, e.g. p2, L.4).

AC: We have replaced the phrase "LGM Holocene dilemma" with "LGM-Holocene transition" and changed "interglacial" to "glacial/interglacial" throughout the manuscript (e.g. Page 30, line 3).

RC: P1, L.22: I am not sure these two references are the best to define the "LGM".

AC: Included (Yokoyama, 2000), ice sheet and glacier proxies (Clark, 2009) and stratigraphic records (Hughes et al, 2013; Hughes and Gibbard 2015) (see P32 L3 of the amended manuscript).

RC: P1, L. 26: and to the fact that the terrestrial carbon content was most likely reduced (e.g. Ciais et al., 2012, Peterson et al., 2014).

AC: we have added the following (P32, L7): "... alongside changes in the terrestrial biosphere stock of carbon (e.g. Francois et al, 1999; Ciais et al, 2012; Peterson et al, 2014; Hoogakker et al, 2016)"

RC: P2, L.2: only the reference to one review (Sigman et al. 2010) is given, while additional references could be given for all the hypotheses cited (at least one per mechanism). Another review could be mentioned: Kohfeld and Ridgwell, 2009.

AC: We have added Kohfeld and Ridgwell (2009), Broecker (1982), Sarmiento and Toggweiler (1984) for the ocean carbon reservoir reference (now on P32 L7).

For the hypotheses cited, we have added (on P32 and P33):

Ocean biology: Martin (1990), Watson et al (2000), Martinez-Garcia (2014) Ocean circulation and mixing/stratification: Toggweiler (1985, 1999), Curry and Oppo (2005, Kohfeld and Ridgewell (2009), Anderson et al (2009), de Boer and Hogg (2014) ), Menviel et al (2016), Muglia et al (2018). Sea ice cover: Stephens and Keeling (2000) Synthesis of mechanisms: Kohfeld and Chase (2017), Ferrari et al (2014). Other features are implicated including temperature, terrestrial biosphere, ocean volume, shelf carbonates. (Trent-Staid and Prell (2002), Annan and Hargreaves (2013), Ciais et al (2012), Opdyke and Walker (1992), Ridgewell et al (2003)).

RC: P2, L.4-11: I would strongly suggest to significantly revise this paragraph, which really does not do justice to the last 15 years of work on the topic of glacial/interglacial changes in atmospheric CO2. Many sensitivity experiments and transient simulations have been performed with box models, models of intermediate complexity and OGCMs to understand glacial/interglacial changes in pCO2. A few references (non-exhaustive list) include Stephens & Keeling (2000), Toggweiler et al., (2006), references within Kohfeld and Ridgewell (2009), Hain et al., (2010), Tagliabue et al., (2010), Hesse et al., (2011), Bouttes et al., (2012), Tschumi et al., (2011), Chikamoto et al., (2012), Menviel et al., (2012), Ganopolski & Brovkin (2017), Menviel et al., (2017). . .. Many of which (if not all of them) also included a thorough model-data comparison.

AC: Paragraph revised, and moved to the modelling section (P32, L18)

RC: On the contrary, I would have liked to see in the introduction more details with respect to the rationale of constructing a new carbon cycle box model.

AC: We have expanded this discussion and added it to the front of the introduction (as per response above; see P2 L16 of the revised manuscript), as well as the discussion section (Section 5).

RC: P2, L. 25: Please reformulate "extra-ocean"

AC: replaced with "carbon cycle" (P2, L34).

RC: (Please also reformulate header of section 2.4)

AC: Replaced with "Atmosphere and terrestrial carbon cycle" (now Section 2.5, P16).

2) Model description

RC: The model description is incomplete. In section 3, it is stated that the model is forced by SST and SSS, however there is no mention of the treatment of temperature and salinity in the model.

AC: We have added a description of the model's treatment of temperature and salinity in Section 2.4 (P15). The temperature and salinity in each of the model's surface ocean boxes is prescribed. The model does not solve for these values, rather takes them as inputs for the calculation of pCO2 in the ocean. We argue that this is a plausible approach for paleo-reconstructions given the emergence of paleo- estimates for SST across glacial-interglacial cycles (e.g. Kohfeld and Chase, 2017), as a useful forcing for model-data exercises.

The starting data are sourced from modern (GLODAPv2) ocean data, mapped into box model space, with adjustments made to the values for the model experiments, e.g. glacial period temperature (decrease) and salinity (increase) are forced. Temperature feeds into the pCO2 / CO2-3 calculation and air-sea fractionation factors for d13C. Salinity feeds into the pCO2 / CO2-3 calculation.

RC: There is no description of the parametrization of the carbon isotopes in the main part of the manuscript. Since the manuscript focuses on carbon isotopes, the main formulations have to be clearly laid out.

AC: We have moved the description of carbon isotopes to the main body of the document (Section 2.7, P17).

RC: In addition, marine export production is prescribed (p9), but there is little information on the values used, how they were chosen and how they vary in the experiment.

AC: We have added more information on the marine export production, as follows (Section 2.2.3 P12, L8):

"We started with a global base productivity/export value at 100m of 5 mol C m-2 yr-1, which falls in the range of Martin et al (1987), of 1.2-7.1 mol C m-2 yr-1 , and Sarmiento and Gruber (2006), 0-5 mol C m-2 yr-1. Additionally, then we have manually tuned the individual surface box values, via a scalar for each box, to match the GLODAPv2 data for DIC, phosphorous, alkalinity, CO2-3 and the carbon isotopes. We chose a value for the "b-scalar" for the Martin et al (1987) export curve, of 0.75, which falls in the range of Berelson et al (2001) of 0.82 +/- 0.16, and also within that of Gloege et al (2017) of 0.68-1.13, and close to the original Martin estimate of 0.858. We found a global value of 0.75 to produce a better fit to the GLODAPv2 data when calibrating the model. The b parameter controls the depth decay of the biological export flux. "

We have added a table (P13, top of page) which shows the initial values for marine export production, and the part of the manuscript dealing with the LGM-Holocene experiments now has a table setting out how the parameters vary in the experiments.

RC: Figure 4 could be helpful in that sense: the late Holocene and/or modern day values of all parameters should be clearly indicated in that figure.

AC: Figure 4 (P 24) amended to include modern day values/assumptions for the parameters shown.

RC: P 3, L. 5-6: "simulates sources and sinks". Some of these sources and sinks are really simplified, for example anthropogenic and volcanic emissions are a simple prescribed flux into the atmosphere. Weathering and river fluxes are also close to a simple pre- scribed flux. So, for some it might be more precise to state "includes forcing" than "simulate sources".

AC: OK, done (Section 2; P2, L30)

RC: P3, L. 13-15; I am confused by this sentence.

AC: Removed offending sentence

RC: Ocean circulation and mixing: Box 4: why is there no exchange with boxes 3, 5 and 7 in equation 1? From the matrix, it looks like there are exchanges with boxes 5 and 7 but not 3, why?

AC: There are a few aspects to this comment. With regard to exchange between Box 4 and Box 3, we have assumed that this flux is small compared with the lateral transport and mixing fluxes between Boxes 4/6 and boxes 1/3. We assume this is the divide between northward flowing water sourced from Antarctic Intermediate Water (AAIW) and Subantarctic Mode Water (SAMW), overlying southward return flow from Atlantic Meridional Overturning Circulation (AMOC) and Pacific/Indian Deep Water (PDW/IDW).

With regard to exchange between Box 4 and boxes 5 and 7, this flux is shown in Equation 4 by the flux (–C4): it is simply a flux out of Box 4. The matrix (Equation 6) shows that this flux is split into Box 7 and Box 5 via the alpha parameter, described in the text.

As general comments on the matrices and the logic of the fluxes. The concentration of an element in each box is a function of a) the magnitude of the physical flux (in Sv) into a box and the element concentration of the originating box and b) the magnitude of the flux (in Sv) out and element concentration of the box itself. The concentration of the 'downstream' box does not enter the equation.

As shown in Figure 1, box 4 receives flux of DIC (C) from box 2 via Psi2. Psi2 also directly transmits to box 7 from box 4, but this is a flux out of box 4 and box 7 does not enter Equation 1. Likewise, Psi1 (red arrow in Figure 1) transmits C from box 6 into box 4 (as per equation 1), but the outward flux of carbon from box 4 into boxes 5 and 7 is function of box 4 element concentration, and boxes 5 and 7 do not need to enter this

equation.

We have added text in the manuscript to specifically address this (P9).

RC: Box 1: why no exchange with boxes 2 & 7 in equation 2?.

AC: Equation 2 refers to the parameter gamma2, which governs mixing between the low latitude surface box (1) and intermediate box (3). We assume that northward lateral transport takes place between the sub polar, intermediate and northern boxes. This water is colder and denser than the overlying mixed layer, given its deep-upwelled sources from AAIW and SAMW from upwelled NADW/PDW/IDW (e.g. Talley, 2013). We assume that Box 1, the low latitude surface box, represents the mixed layer (e.g. Kara et al, 2013), which is mainly under the influence of ocean surface processes. We prescribe vertical mixing between this box and the underlying intermediate box via the gamma2 parameter, conceptually the thermocline mixing described by Liu et al (2016).

As such, the parameter only operates on boxes 1 and 3 as per equation 2 (and as shown in Figure 1).

We have added text in the manuscript to specifically address this (P10).

RC: P11, L. 15-17: "around glacial cycles" is not precise enough. In addition, I don't think this sentence is correct, as changes were opposite in the Atlantic and Pacific Oceans.

AC: (P13, L20) reworded as: "it is a dynamic process, and the dissolution and burial in sediments of CaCO3 is observed to vary across (and within) glacial/interglacial cycles), suggesting an influence on carbon cycling".

The aim of this sentence is to briefly introduce carbonate sediment burial and dissolution as an influence on the carbon cycle.

3) Modelling results

RC: P16, L. 17: Is [CO3] approximated by ALK-DIC or fully calculated using ALk, DIC,

T, S, P?

AC: The latter. We use the method of Follows et al (2006) which calculates pCO2 and CO2-3 as a function of Alk, DIC, T, S and P. The purpose of this sentence was to highlight that the approximation for CO2-3 of Alk-DIC is useful for interpreting model results charts. We have amended this sentence accordingly (P22 L23), and expanded the description of the pCO2 and carbonate ion calculations to identify DIC, Alk, T, S and P as inputs (P13 L10).

RC: P16, L.20: please reformulate as "remineralization of organic matter"

AC: Amended (P22, L25).

RC: P17, L. 2-7: Please explain your reasoning behind varying the rain ratio.

AC: This paragraph has been re-worded, with the first reference to the rain ratio removed – as it is confusing (P23, L12).

RC: I don't understand why changing the rain ratio impacts atmospheric D14C and I suppose that the surface ocean pCO2 change could eventually impact atm d13C, but not "heavily" (L. 6-7).

AC: Re atmospheric D14C. Increasing the rain ratio leads to higher pCO2 in the ocean surface boxes (removes alkalinity in ratio 2:1 to DIC), and subsequent de-gassing of CO2 to the atmosphere, which increases atmospheric CO2. The air-sea fractionation factors for D14C, that we have used, exhibit greater fractionation of the isotopic ratio in out-gassing to the atmosphere versus in-gassing to the ocean, so there is a modest decrease in atmospheric D14C (the atmosphere is preferentially receiving 12C). We have removed the word "heavily" as that wording indeed exaggerates the effects (P23, L26).

RC: P19 L6. Please add "and there is a reduced outgassing of old low D14C waters"

AC: Amended accordingly (P25, L3).

RC: P19 L8. Please remove "around the interglacial cycles" and please note that the year of the reference is actually 2008.

AC: Done, reference updated throughout document (e.g. P30, L12).

P19, L. 9-14: I suppose the authors expect a change in pCO2 due to the change in ocean area resulting from varying sea-level (and thus ocean volume) on G/IG timescales. Please spell it out. Please take out "volume" on L.9. The impact on D14C is surprising though.

AC: Amended accordingly (P25, L10).

RC: P22 L7. This sentence is not correct, re-formulate.

AC: We recompiled this section as part of the discussion of LGM-Holocene work (Section 4, P30)

RC: P22, L. 10-14: I don't really agree with this paragraph. It is probably true for simple carbon cycle box model for which all parameters have to be tested and therefore the G/IG CO2 problem is explored by assessing the impact of each parameters. But, over the last years the G/IG CO2 problem has also been studied with coupled models providing a representation (granted this representation is associated with large uncertainties) of physical and bio- logical changes occurring during glacial times.

AC: Easier to remove this paragraph.

RC: P23 L4-6 "I am not sure what is meant here or what has been done"

AC: Sentenced removed as is extraneous.

4) Discussion

RC: A discussion of the capacity of the model and the results is missing. I would have liked to see a paragraph on why this model should be used. What are its benefits and limitations? Its fast processing time should be discussed here, instead of the introduction. I would have liked to see the results of the future experiments discussed in the context of the CMIP5 results. Only Jones et al and Wang are referenced in this part. I would have liked to see a discussion of the results of the LGM experiments in comparison with other studies. Recently Muglia et al (2018) and AMOC, iron fertilisation. Menviel was consistent. These two studies among others could help discuss the effects of Z, psi1, psi2 as shown in Figure 10.

AC: We have added new discussion Sections 5.1 (Model advantages and limitations), 5.2 (Modern carbon cycle simulations) and 5.3 (LGM-late Holocene modelling) to address this comment.

RC: Abstract The second part of the abstract focusses on the LGM simulations. I would suggest to tone down that part and instead add some information about the use and limitations of the model.

AC: Noted and amended accordingly, incorporating summary of limitations described above.

RC: Minor and typos P3 L10-12 please reformulate AC: Easier to remove, as the point is made in the preceding sentences P16 L29-35. Please reformulate this paragraph. AC: Amended (P23, L11-24) P17 L10 "decreases" AC: Fixed (P23, L30) P19 L19 please reformulate sentence. AC: Amended (P25, L24) P19 L26. Maybe "appropriate" instead of "accurate". AC: Fixed (P25, L32)

Please also note the supplement to this comment:
https://www.geosci-model-dev-discuss.net/gmd-2018-176/gmd-2018-176-AC1-supplement.pdf
* * *
[Figure]

**Supplement:**

[revised manuscript text omitted]

---

## Author Comment (AC2) · 6 Feb 2019

Thank you for your constructive and thorough comments, suggestions and input into the manuscript. We feel it makes a very strong contribution to the quality of the work. Please see below our responses to the individual comments. We have made reference to changes to the manuscript, which is included as a supplement to the author comments, in track changes.

Page and line references below refer to locations in the revised document with track changes. Please note the attached, marked-up document contains amendments from both sets of reviewer comments.

RC: This paper is mostly well-written. The model is nicely described and is a genuinely useful framework for investigating physical and biogeochemical controls on the marine carbon cycle. I have no serious concerns with the work and my comments are mostly

suggestions for rewording and clarification. A few slightly more important issues are below, followed by line-by-line notes.

RC: I did not see a description of the numerical method employed or confirmation of model stability and potential for numerical error. Fig 7 shows some potential error propagation / numerical oscillation? Have the authors investigated this?

AC: the model equations are a set of partial differential equations, one for each element in the model. These are solved with a straightforward 1st order Euler forward time-stepping method with a standard timestep of one year. We find the model to be stable and approaching steady state for most of the simulations we have undertaken. However, as noted by the reviewer, this stability is challenged by scenarios with strong forcing.

Figure 7 of our original submission shows this instability for the extreme emissions scenario RCP8.5. We have re-run this scenario with a reduced timestep (0.5 years) and find that the weak instability in the model results for air-sea carbon exchange, is eliminated. We have also run the other RCP scenarios at reduced timestep, which shows a smoother trajectory for air-sea gas exchange of carbon, shown in the revised Figure 7.

We have added a description of the numerical method (P8, L6).

RC: Figure placement needs significant improvement. For example, Figure 9 appears 5 pages after it is mentioned and is in a different section.

AC: We have revised the figure and table placements. Note some of these are slightly out of place in the marked-up version of the manuscript, but this is resolved in the clean document.

RC. A few times it is noted that the model supports a physical overturning mechanism for driving LGM-Holocene changes, it should be made clearer that this idea has been proposed before and the current work supports it, rather than introducing the concept.

AC: In the discussion of the LGM-Holocene modelling results, we have made mention of previous findings. This point is addressed specifically in Section 5.3 (P42, L11).

Line-by-line:

Page2: line 15: summary of box models is too vague

AC: We have expanded this summary in the introduction (P2, L16)

Page 2, Line 32: "simple carbon project model model"

AC: offending duplicate removed

RC: Page 3: Line 20: does 'zonally averaged' make sense? There are no spatial dimensions here? E.g. later zonally-averaged refers to a 2D model.

AC: We agree that the use of "zonally averaged" when referring to box models and the Talley 2-D conceptual model is confusing. We have removed "zonally averaged" from both instances (P5, L4; P6, L7).

RC: Figure 1. It is not always clear which arrows exchange with which boxes. E.g. some arrows are entirely within one box, some cross box boarders but do not terminate. The diagram should show what is actually happening in the model.

AC: we assume this comment refers to the red and orange arrows. We have amended Figure 1 by trimming the arrows to show only where there is a flux between boxes, via shortened arrows that cross the relevant box border, and removing arrows wholly contained within box borders.

RC: Page 5 Line 14 to end: A little confusion over model dimensionality. Be precise here. Explain how the model has no spatial dimensions but does have a representation of sizes and locations of boxes (if that is indeed correct).

AC: We have clarified the relationship between model boxes and spatial dimensions in the real ocean (P6, L16).

[Figure]

RC: Figure 2. Caption – "implemented".

AC: Amended

RC: Figure 2. Also explain the direction of arrows here.

AC: We have added the following statement to the caption for Figure 2: "The arrows refer to the direction of file linkages and the order of their activation during the routine of setting up and running the model."

RC: Page 9: Line 1: Biological flux "parameters" or "parameterization".

AC: now "Biological flux parameterisation"

RC: Line 2: "action of biological activity" – reword

AC: On page 10 last paragraph (L12), reworded as: "The biological pump (e.g. Broecker, 1982) is a descriptor of marine biological activity, whereby organisms consume nutrients in shallow waters, die, sink and then release those nutrients at depth."

RC: Line 21: sub-surface or subsurface

AC: sub surface replaced with "subsurface" throughout (e.g. P11, L17).

RC: Page 11: Line17: 'lending it some interest', consider rewording

AC: On page 13 Section 2.3.1 reworded as: "According to Farrell and Prell (1989) it is a dynamic process, and the dissolution and burial in sediments of CaCO3 is observed to vary across (and within) glacial/interglacial cycles), suggesting an influence on carbon cycling".

RC: Line 34: should be "non-saturation-dependent" ?, "Earth"

AC: Amended as such (P14, L21-22)

RC: Page 13: line 5: replace "extra ocean" with something more descriptive

AC: Section 2.5, page 16 we have replaced with "Atmosphere and terrestrial carbon

cycle"

RC: line 22: non- zero

AC: Amended (P16, L25)

RC: line 25: short-term, long-term

AC: Amended (P16 and P17)

RC: Page 14: Line 3: state meaning of beta parameter

AC: added to page 17, line 6: "beta is the parameterisation of carbon fertilisation, causing NPP to increase (decrease) logarithmically with rising (falling) atmospheric CO2 levels, with a typical value of 0.4-0.8 (Harman et al, 2011)."

RC: Figure 4: This is hard to read, consider better ways to display (e.g. title and unit on x axis)

AC: We have revised Figure 4 by reducing the number of subplots shown, as the points can be made with one chart for each of sea surface temperature, salinity and piston velocity. We have also consolidated the y-axes labels to reduce clutter. The subplots now include the modern-day values/assumptions used in the model (P24).

RC: Page 19: Line 7: "the carbon isotopes" reword

AC: Page 25, line 6. Reworded as "fractionates the ratios of the isotopes of carbon leading to higher values for d13C and to a lesser extent, D14C, in the atmosphere".

RC: Line 12 "the values for the isotopes" too vague

AC: Page 25, line 12 reworded as: "Increasing the fraction of deep water upwelled into the sub polar surface ocean box (Fig. 4(l)) intuitively raises CO2, but lowers d13C and D14C, by upwelling carbon rich and isotopically-depleted water to the ocean surface boxes."

RC: Line 17: "response to the shocks" does not give the right impression

AC: We agree, and have altered this sentence to read (P25, L18): "In response, the Earth's carbon cycle continually partitions carbon between its component reservoirs, with positive and negative feedbacks. The net effect is a build-up of carbon in most reservoirs".

RC: Figure 5. Explain data in panel b, why are there two data lines for the atmosphere?

AC: this was an issue with the Atmospheric D14C data we have gathered, which has now been rectified in Figure 5 (P26).

RC: Figure 5. Remove "selection of boxes shown to reduce clutter", ironically this statement is itself clutter.

AC: amended as suggested (P26)

RC: Figure 6. Remove "fed into" from caption. Use inputted or similar.

AC: reworded as "…….which are inputted to SCP-M for the modern carbon cycle simulation." (P28)

RC: Page 20: Line 17: "carbon cycle destination for human emissions" – not clear what this means

AC: P27, 33: "Figure 8 shows the partitioning of anthropogenic CO2 emissions into the carbon cycle reservoirs by 2100, as simulated with SCP-M."

RC: Line 19-20: explain this in more detail, a little confusing

AC: we think it is better to delete this sentence altogether. The point is a bit nuanced and perhaps extraneous. We have changed the chart to show a slightly different metric so the sentence is redundant.

RC: Line 22: the figures are becoming a long way from the relevant text by this point in the paper.

AC: Modified in the manuscript throughout. Note that some of the figures and tables

are slightly out of place in the marked-up response attached as a supplement (due to the presence of deleted text), however it is resolved in the clean document (without track changes).

RC: Figure 7: use of multiple transparencies and colors here makes it very hard to see the ranges, especially in the bottom panel. Also, it appears there is some oscillation developing in the model? Have you investigated this?

AC: Yes, for the extreme emissions scenario RCP 8.5 there is some numerical oscillation at the end of the simulation. Please see second AC above for our response. The simulations have been run at reduced step size and the model result trajectories are smoother. The Figure is revised on page 29.

With regards to the transparencies for the CMIP-5 model ranges in Figure 7, we have employed a mixture of hatching and infill, to better distinguish the ranges (P29).

RC: Figure 8: This is quite simple, can a comparison be incorporated?

AC: We have incorporated the corresponding model results from the IPCC WG1 5th assessment report (Chapter 6) in pie chart for comparison, and referenced in Figure caption (P30).

RC: Line 22: "release of emissions" should be reworded

AC: P28 L3: Reworded as: "By 2100 in RCP 6.0, the carbon cycle is substantially changed from the preindustrial/late Holocene state as a result of the accumulation of hundreds of years of human industrial $CO_2$ emissions (Fig. 9)."

RC: Figure 9: I would consider if there is a better way to show this. It takes a very long time to decode this information. Bar or pie charts would be more easily understood.

AC: we acknowledge the comment and agree this is a busy figure. However, we feel that Figure 9, and the information it shows on carbon fluxes between boxes in the model, is intrinsic to the model documentation. This is because it displays flux values

which we believe are plausible, and that this demonstrates the validity of the model for this purpose (modelling carbon fluxes between the different Earth reservoirs). To help simplify, we have replaced the absolute values for the scenario results (the RCP6.0) with the + or – variation from the baseline scenario, in PgC, to highlight what is changing.

RC: Page 22: Line 8: first line here needs clarification.

AC: This section (Section 4, P30) has been substantially revised.

RC: Page 25: Note that table 4 is in the appendix

AC: amended to "as per Table 6 in the Appendix" (P33, L18).

RC: Page 29 Line 8: net respiration versus net uptake should be made clear

AC: Page 37, line 5: amended as "…….effect of the terrestrial biosphere, which causes net uptake CO2 in the Holocene period (increases atmospheric $\delta$13C), and net respiration of CO2 in the LGM period (decreases atmospheric $\delta$13C)".

RC: Page 30 Line 10: don't need 'however' here

AC: Amended (P37, L18): "SCP-M results shown for comparison with sparse carbonate proxy data"

RC: Line 30: "showed deltas in the range of" – be more precise here

AC: We have expanded to (P40, L10): "According to Francois et al. (1999), palynological and sedimentological data infer that the terrestrial biosphere carbon stock was 700-1350 PgC smaller in the LGM, than the present."

RC: Page 31 Line 6: "but critically *are* accompanied"

AC: amended accordingly (P42, L9)

RC: Line 7: *the* carbonate ion proxy?

AC: amended accordingly (P42, L11)

RC: Line 23: superscript 14C

AC: amended (P42, L12; P43, L17)

RC: Line 25: it cannot be wholly explained by overturning changes, as these must be combined with temp/salinity etc. changes listed afterwards. Also this work confirms that reduced overturning can drive the changes, rather than proposing this.

AC: "wholly" has been removed (P43, L20). Page 42, line 11 specifically addresses the second part of this comment. "This is not a new finding. . ..." etc.

RC: Figure 12: as with Figure 9, it's very difficult to get any meaning from this figure. Perhaps colours could be used to denote increases/decreases at the very least?

AC: We have modified the figure to show the positive or negative variation for the LGM, in PgC, to highlight what is changing, as with Figure 9.

Please also note the supplement to this comment:
https://www.geosci-model-dev-discuss.net/gmd-2018-176/gmd-2018-176-AC2-supplement.pdf

---

## Author Response (AR2)

Dear Paul,

Many thanks for your Topical Editor comments on the revised manuscript for "The [simple carbon project] model v1.0". We followed your suggestion and undertook another round of editing the manuscript, involving the full team of authors. We have made minor edits throughout the document, as shown in the track changes version below. Please note that the track changes are with reference to the most recent version of the manuscript, post revision with Referee Comments. The focus of our edits was to improve logic, readability and flow, and to clean-up any remaining typos.

As suggested, we have revised Section 5 to evidence key points with respect to sections, figures and tables in the manuscript, and additional external references as necessary.

We think the manuscript is substantially improved from these changes, and thank you for the input.

Best regards
Cameron and team

[revised manuscript text omitted]